# The structure and organization of lanceolate mechanosensory complexes at mouse hair follicles

Lishi Li[1‡], David D Ginty[1,2*]

[1]The Solomon H Snyder Department of Neuroscience, Howard Hughes Medical Institute, The Johns Hopkins University School of Medicine, Baltimore, United States; [2]Department of Neurobiology, Howard Hughes Medical Institute, Harvard Medical School, Boston, United States

**Abstract** In mouse hairy skin, lanceolate complexes associated with three types of hair follicles, guard, awl/auchene and zigzag, serve as mechanosensory end organs. These structures are formed by unique combinations of low-threshold mechanoreceptors (LTMRs), Aβ RA-LTMRs, Aδ-LTMRs, and C-LTMRs, and their associated terminal Schwann cells (TSCs). In this study, we investigated the organization, ultrastructure, and maintenance of longitudinal lanceolate complexes at each hair follicle subtype. We found that TSC processes at hair follicles are tiled and that individual TSCs host axonal endings of more than one LTMR subtype. Electron microscopic analyses revealed unique ultrastructural features of lanceolate complexes that are proposed to underlie mechanotransduction. Moreover, Schwann cell ablation leads to loss of LTMR terminals at hair follicles while, in contrast, TSCs remain associated with hair follicles following skin denervation in adult mice and, remarkably, become re-associated with newly formed axons, indicating a TSC-dependence of lanceolate complex maintenance and regeneration in adults.

**\*For correspondence:**
david_ginty@hms.harvard.edu

**‡Present address:** Laboratory of Mammalian Cell Biology and Development, Howard Hughes Medical Institute, The Rockefeller University, New York, United States

**Competing interests:** The authors declare that no competing interests exist.

**Reviewing editor**: Jeremy Nathans, Howard Hughes Medical Institute, Johns Hopkins University School of Medicine, United States

## Introduction

Our skin is the largest sensory organ of the body and is presented with an array of tactile stimuli, including indentation, stretch, vibration, and hair deflection (*Lumpkin et al., 2010*). To detect, perceive, and respond to such diverse stimuli, morphologically and physiologically distinct classes of low-threshold mechanosensory neurons (LTMRs) innervate the skin and associate with cutaneous tactile end organs. These LTMRs, whose cell bodies are located in dorsal root ganglia (DRG) and cranial sensory ganglia, carry impulses from their endings in the skin to the central nervous system (*Abraira and Ginty, 2013*; *Lechner and Lewin, 2013*). LTMRs are classified as Aβ, Aδ or C based on their action potential conduction velocities (*Horch et al., 1977*; *Rice and Albrecht, 2008*). C-LTMRs are unmyelinated and thus have the slowest conduction velocities, whereas Aδ-LTMRs and Aβ-LTMRs are lightly and heavily myelinated, exhibiting intermediate and rapid conduction velocities, respectively. LTMRs are also classified as slowly, intermediately, or rapidly adapting (SA, IA, and RA-LTMRs) according to their rates of adaptation to sustained mechanical stimuli (*Burgess et al., 1968*; *Johnson and Hsiao, 1992*). Genetic labeling of Aβ RA-LTMRs, Aδ-LTMRs, and C-LTMRs has revealed that the cutaneous endings of each LTMR subtype in hairy skin form longitudinal lanceolate endings at one or more hair follicle subtypes (*Li et al., 2011*).

More than 90% of the body surface of most mammals is covered by hair follicles, which regulate body temperature, facilitate perspiration, and are involved in the perception of mechanosensory stimuli upon the skin. There are three major hair follicle subtypes in mouse back hairy skin: guard, awl/auchene, and zigzag. The three hair follicle subtypes emerge during different stages of hair follicle

**eLife digest** Many mammals, such as cats, mice, and sea lions, have long whiskers that are particularly sensitive to touch. However, the hairs that cover the skin of most mammals are also important touch detectors. These hairs grow from follicles that are connected to the ends of the nerve cells that detect and convey touch information to the central nervous system. In mice, three main types of hair follicle—guard hairs, awl hairs, and zigzag hairs—are associated with combinations of three types of nerve endings. Much remains to be understood about how hair follicles and nerve cell endings—which are wrapped by cells called terminal Schwann cells—interact via structures called lanceolate complexes.

Now, using a combination of genetics, microscopy and surgical procedures, Li and Ginty have studied these structures in unprecedented detail, and revealed some intriguing structural differences among the three types of hair follicles. Zigzag follicles—which make up the fur undercoat—are associated with fewer terminal Schwann cells than are awl follicles, whilst guard hair follicles have the most. High-resolution analyses revealed that distinct combinations of sensory nerve endings were associated with different types of hair follicle cells—which may underlie the unique responses of the different hair follicle types when the hairs are deflected. Furthermore, an individual terminal Schwann cell can be associated with more than one type of nerve ending, adding another layer of intricacy to the detection of hair movements.

Killing the terminal Schwann cells in mice caused a complete loss of sensory nerve endings at hair follicles, which suggests that these cells are essential for maintaining the connection between the hair follicles and nerve cell endings. Interestingly, surgically removing nerve endings from the skin did not lead to a loss of terminal Schwann cells, and the nerve endings eventually grew back and reconnected with the hair follicles.

In addition to shedding new light on the structures of lanceolate complexes in different types of hair follicles, the findings of Li and Ginty suggest that terminal Schwann cells maintain the nerve endings at hair follicles and guide their regeneration after damage. Uncovering the molecular mechanisms that control these processes represents an important next step in this research.

morphogenesis and have distinct morphologies and presumably distinct mechanical properties. Guard hairs, which make up ~1% of the back skin hairs of the mouse, develop during the first wave of hair follicle morphogenesis, beginning E13–14.5. These hairs have the longest hair shafts that have two rows of medulla cells and do not contain kinks. Awl hairs, which form during the second wave of murine hair morphogenesis, beginning E15–16.5, are straight but are shorter than guard hairs and contain three or four rows of medulla cells. Auchene hairs are developmentally and morphologically identical to awl hairs, except that they have one kink in the hair. Awl/auchene hairs together make up ~20% of back skin hairs and, for the purposes of this study, are considered together. Finally, zigzag hairs form during the third wave of hair morphogenesis beginning at E18.5 and are the most abundant type, making up ~80% of hairs. Zigzag hairs have one row of medulla cells and at least two bends in their hair shaft (*Schlake, 2007*; *Driskell et al., 2009*; *Li et al., 2011*). In addition to their different developmental and morphological properties, zigzag, awl/auchene, and guard hair follicles are each associated with unique combinations of LTMRs. Indeed, guard hair follicles are innervated by Aβ RA-LTMR lanceolate endings and are also associated with Aβ SA1-LTMR Merkel endings. In contrast, awl/auchene hairs are triply innervated by inter-digitated Aβ RA-, Aδ-, and C-LTMR lanceolate endings, whereas zigzag hair follicles are innervated by inter-digitated Aδ- and C-LTMR lanceolate endings (*Li et al., 2011*). Thus, guard, awl/auchene, and zigzag hairs, with their unique combinations of LTMR endings, play neurophysiologically distinct roles in mechanosensory transduction (*Iggo and Andres, 1982*; *Li et al., 2011*).

Light and electron microscopic studies have shown that longitudinal lanceolate endings are arranged parallel to the hair follicle long axis and that each ending is encased in finger-shaped terminal Schwann cell (TSC) processes (*Yamamoto, 1966*; *Halata, 1993*; *Kaidoh and Inoue, 2000*; *Takahashi-Iwanaga, 2000*). Furthermore, intercellular junctions are observed between axon terminals, TSCs, and hair follicle outer root sheath cells (*Kaidoh and Inoue, 2000, 2008*), suggesting that physical interaction between

these different cell types may be essential for mechanotransduction. However, the organization of TSCs and LTMR lanceolate endings, and their functional relationship for maintenance of lanceolate complexes are largely unknown. In this study, we used histologic and genetic approaches to investigate the functional organization, ultrastructural properties, and maintenance of lanceolate complexes associated with the three hair follicle subtypes of the mouse.

## Results

### The organization of lanceolate complexes at guard, awl/auchene, and zigzag hair follicles

We previously reported molecular–genetic strategies that enable visualization of axonal endings of C-, Aδ-, and Aβ RA-LTMRs in the skin and spinal cord of the mouse. We found that C-LTMRs can be visualized in $Th^{CreER}$; $Rosa26^{LSL-tdTomato}$ mice treated with 4-HT; Aδ-LTMRs are visualized using $TrkB(Nrtk2)^{tauEGFP}$ knockin mice; and Aβ RA-LTMRs are labeled in *Npy2r-GFP* BAC transgenic mice (*Li et al., 2011*). Remarkably, in back hairy skin of the mouse, C-, Aδ-, and Aβ RA-LTMRs form morphologically similar longitudinal lanceolate endings associated with hair follicles (*Figure 1*). Immunostaining with S100, a glial cell marker, shows that each type of LTMR lanceolate ending is associated with TSCs, with cell bodies of TSCs residing at the base of lanceolate complexes. Moreover, TSCs associated with C-, Aδ-, and Aβ RA-LTMRs exhibit extensive longitudinal processes extending toward the skin surface. These finger-shaped processes are parallel to the long axis of the hair follicle and encase individual longitudinal lanceolate endings of C-, Aδ-, and Aβ RA-LTMRs (*Figure 1*). These observations are consistent with previous findings from our group and others (*Iggo and Andres, 1982*; *Li et al., 2011*; *Woo et al., 2012*).

Whether lanceolate complexes comprised of unique combinations of LTMR subtypes around the three types of hair follicles exhibit unique morphological, ultrastructural, or organizational patterns is an intriguing, unanswered question. Indeed, whole-mount immunostaining of back hairy skin using anti-S100 shows that lanceolate complexes associated with the three different hair follicle subtypes appear to contain different numbers of TSCs (*Figure 2A,B*). Quantification of TSCs associated with individual hair follicles shows that individual awl/auchene hair follicles have slightly more TSCs compared with zigzag hair follicles, and that guard hair follicles have many more TSCs than both zigzag and awl/auchene hairs (*Figure 2C*). Since zigzag, awl/auchene, and guard hair follicles have different

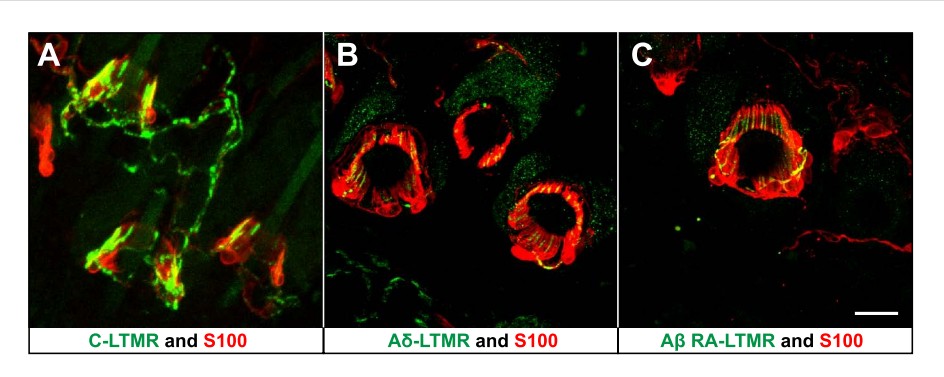

**Figure 1**. LTMRs and TSCs form palisade-like lanceolate complexes at hair follicles. (**A**) On back hairy skin sections from $Th^{CreER}$;$Rosa26^{tdTomato}$ mice, C-LTMRs are visualized using tdTomato fluorescence (green), while TSCs are labeled using S100 immunostaining (red). C-LTMRs form longitudinal lanceolate endings associated with TSCs at zigzag and awl/auchene hair follicles. (**B**) Back hairy skin sections from $TrkB^{tauEGFP}$ animals were stained with anti-GFP to label Aδ-LTMR axonal terminals and anti-S100 (red) to label TSCs. Aδ-LTMRs form longitudinal lanceolate endings associated with TSCs at zigzag and awl/auchene hair follicles. (**C**) Back hairy skin sections from *Npy2r-GFP* animals were stained with anti-GFP to label Aβ RA-LTMR axonal terminals and anti-S100 (red) to label TSCs. At a representative awl/auchene hair follicle, Aβ RA-LTMRs form longitudinal lanceolate endings associated with TSCs. Similar patterns can be seen at guard hair follicles (*Li et al., 2011*). Animals around 3 weeks of age were used for these experiments. Scale bar, 20 μm.

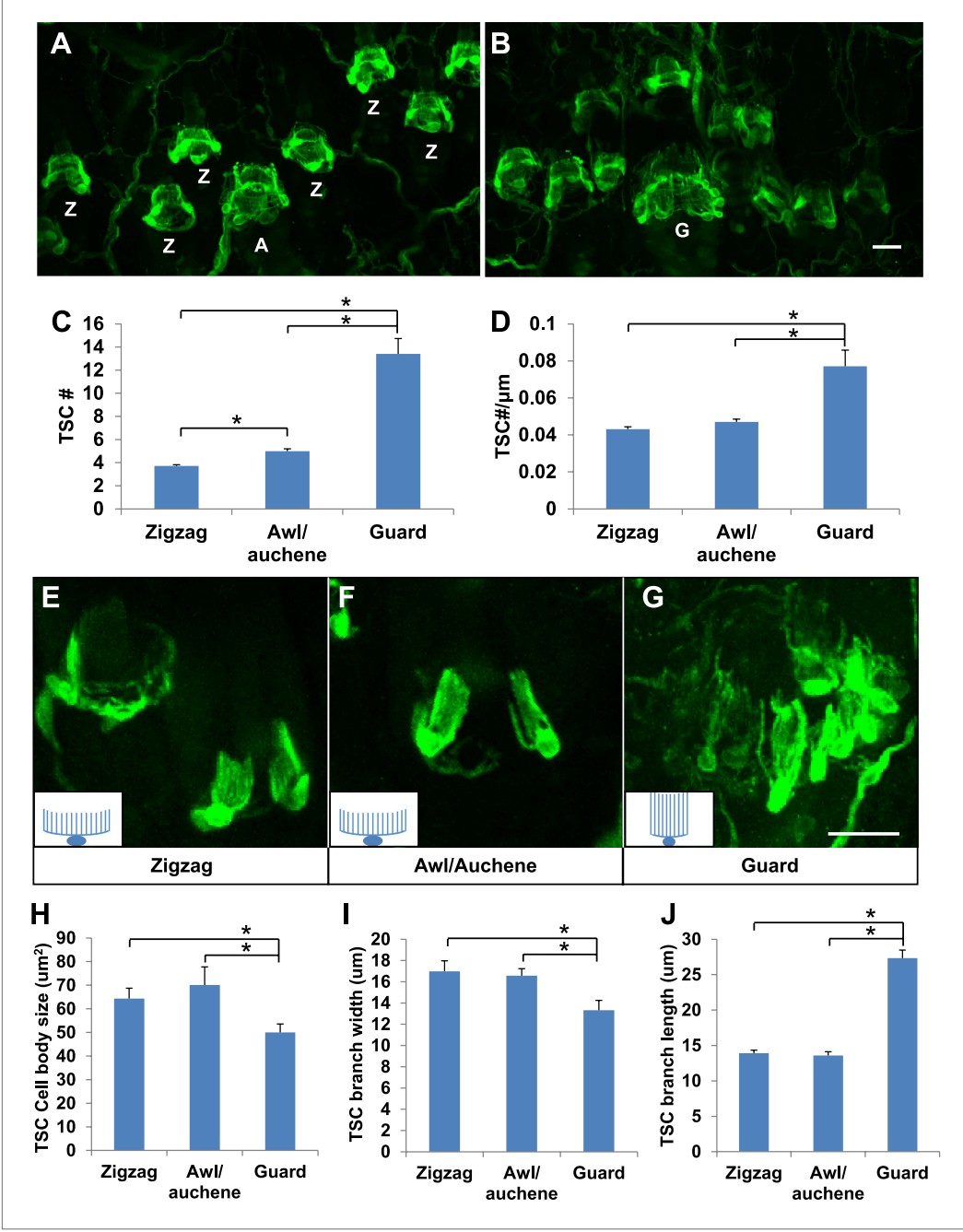

**Figure 2**. Guard, awl/auchene, and zigzag hair follicles have different numbers and morphologies of TSCs. (**A** and **B**) Whole-mount immunostaining of back hairy skin using anti-S100 shows that awl/auchene (**A**) and zigzag (**Z**) hair follicles (in panel **A**) and guard (**G**) hair follicles (in panel **B**) have different numbers of TSCs. Scale bar, 20 μm. (**C**) Comparisons of numbers of TSCs at individual hair follicles. Awl/auchene hairs (5.0 ± 0.2 TSCs, n = 24 hair follicles) have slightly more TSCs than zigzag hairs (3.7 ± 0.1 TSCs, n = 64 hair follicles) (p<0.001). Guard hairs (13.4 ± 1.3 TSCs, n = 5 hair follicles) have many more TSCs than awl/auchene and zigzag hairs (p<0.001 for both comparisons). (**D**) Densities of TSCs at individual hair follicles were calculated by dividing the number of TSCs by the circumference of the hair follicles. Although TSC densities are comparable between zigzag (4.3 ± 0.1 TSCs/ 100 μm) and awl/auchene hairs (4.7 ± 0.2 TSCs/100 μm) (p=0.084), guard hair follicles (7.7 ± 0.9 TSCs/100 μm) have almost twofold higher densities of TSCs compared to zigzag and awl/auchene hair follicles (p<0.001 for both comparisons with zigzag and awl/auchene hair follicles). (**E**–**G**). In *Plp1^CreER;Rosa26^YFP* animals treated with 0.01 mg of tamoxifen, TSCs were sparsely labeled to visualize the morphologies of individual TSC associated with different

*Figure 2. Continued on next page*

*Figure 2. Continued*

hair follicle types. Insets in **E–G** are schematic images of TSCs to summarize the differences observed between TSCs associated with guard hairs vs awl/auchene and zigzag hairs. Scale bar, 20 µm. (**H–J**) Sizes of TSC cell bodies (**H**), widths of total processes of individual TSCs (**I**) and lengths of TSC processes (**J**) were measured at zigzag, awl/auchene, and guard hair follicles. Although all three parameters are comparable between zigzag and awl/auchene hair follicles, indicating similar morphologies, individual TSCs at guard hair follicles have significantly smaller cell sizes, and narrower and longer processes compared to the other two hair follicle types (Zigzag hair follicles: cell body size, $64.3 \pm 4.5$ µm$^2$; width of processes, $17.0 \pm 1.0$ µm; length of processes, $13.9 \pm 0.4$ µm; n = 25 hair follicles. Awl/auchene: cell body size, $70.1 \pm 7.6$ µm$^2$; width of processes, $16.6 \pm 0.7$ µm; length of processes, $13.6 \pm 0.6$ µm; n = 10 hair follicles. Guard hair follicles: cell body size, $50.0 \pm 3.5$ µm$^2$, $p<0.05$ compared with zigzag or awl/auchene hairs; width of processes, $13.3 \pm 0.9$ µm, $p<0.05$ compared with zigzag or awl/auchene hairs; length of processes, $27.3 \pm 1.1$ µm, $p<0.001$ compared with zigzag or awl/auchene hairs; n = 16 hair follicles). Animals around 3 weeks of age were used in these whole-mount immunostaining experiments.

diameters, we also compared the densities of TSCs at lanceolate complexes by dividing the number of TSCs by the circumference of individual hair follicles. Interestingly, TSC densities for lanceolate complexes associated with zigzag and awl/auchene hair follicles are comparable, while TSCs at guard hair follicles are considerably more compact (*Figure 2D*). We next asked whether individual TSCs at zigzag, awl/auchene, and guard hair follicles have similar or distinct morphologies. To visualize the morphological properties of individual TSCs at the three hair follicle types, we crossed the *Plp1$^{CreER}$* mouse line, which expresses CreER exclusively in glial cells (*Doerflinger et al., 2003*), to a *Rosa26$^{YFP}$* reporter mouse line, and achieved sparse labeling of TSCs by treating double transgenic animals with a single, low dose of tamoxifen (0.01 mg–0.03 mg; *Figure 2E–G*). Whole-mount immunostaining with anti-GFP allowed visualization of individual TSCs and measurement of the size of TSC cell bodies, as well as the width and length of their processes (*Figure 2E–G,H–J*). We found that the morphologies of individual TSCs associated with zigzag and awl/auchene hair follicles are comparable. In contrast, TSCs at guard hair follicles have smaller cell bodies, and fewer and considerably longer processes (*Figure 2H–J*). The greater numbers and unique morphologies of TSCs associated with guard hair follicles compared to those associated with zigzag and awl/auchene follicles, together with the finding that Aβ RA-LTMRs represent the sole longitudinal lanceolate endings at guard hairs, suggests unique mechanotranduction functions of guard hairs in detecting hair movements.

Three and two distinct LTMR subtypes form lanceolate endings associated with most if not all awl/auchene and zigzag hair follicles, respectively (*Li et al., 2011*). Consistent with this, recent studies indicate that mechanosensory neurons with distinct molecular features can innervate the same hair follicle (*Heidenreich et al., 2012*; *Wende et al., 2012*; *Woo et al., 2012*). These observations raise the intriguing question of whether physiologically distinct LTMR subtypes that innervate the same hair follicle associate with the same or different TSCs. Based on the interdigitated patterns of LTMR axonal endings at zigzag and awl/auchene hair follicles (*Li et al., 2011*), there are at least two possible organizational patterns of their associated lanceolate complexes: 1) Each TSC hosts only one type of LTMR, and the interdigitated axonal terminals of different LTMRs are associated with interdigitated processes of TSCs; 2) Processes of TSCs at individual hair follicles are tiled and each TSC hosts interdigitated axonal terminals from two or more LTMR subtypes. To distinguish between these possibilities, we first crossed the *Plp1$^{CreER}$* mouse line to the multicolor Cre-reporter *Rosa26-Confetti* (*Schepers et al., 2012*) to label TSCs with multiple reporters. Tamoxifen treatment of *Plp1$^{CreER}$;Rosa26-Confetti* mice allows individual TSCs to express only one of the four fluorescent proteins encoded by the *Rosa26-Confetti* allele. Thus, adjacent TSCs will express different reporters allowing visualization of the organization of TSCs and their associated LTMR endings at individual hair follicles. In trunk hairy skin of *Plp1$^{CreER}$;Rosa26-Confetti* animals, adjacent TSCs that are labeled by red and green fluorescence reporters exhibit non-overlapping sets of processes (*Figure 3A,B*). Thus, the processes of TSCs associated with individual hair follicles are tiled. Moreover, this finding indicates that the organization of individually tiled TSCs and different LTMR endings can be visualized simply by immunostaining with glial cell markers. Interestingly, in back hairy skin sections from *Th$^{CreER}$;Rosa26$^{tdTomato}$;TrkB$^{tauEGFP}$* mice, in which C-LTMRs, Aδ-LTMRs, and TSCs are visualized with tdTomato, GFP, and S100 immunostaining, respectively, we found that C-LTMRs and Aδ-LTMRs form interdigitated lanceolate endings associated with processes belonging to the same TSC (*Figure 3C,C'*). Similar patterns were observed in skin sections

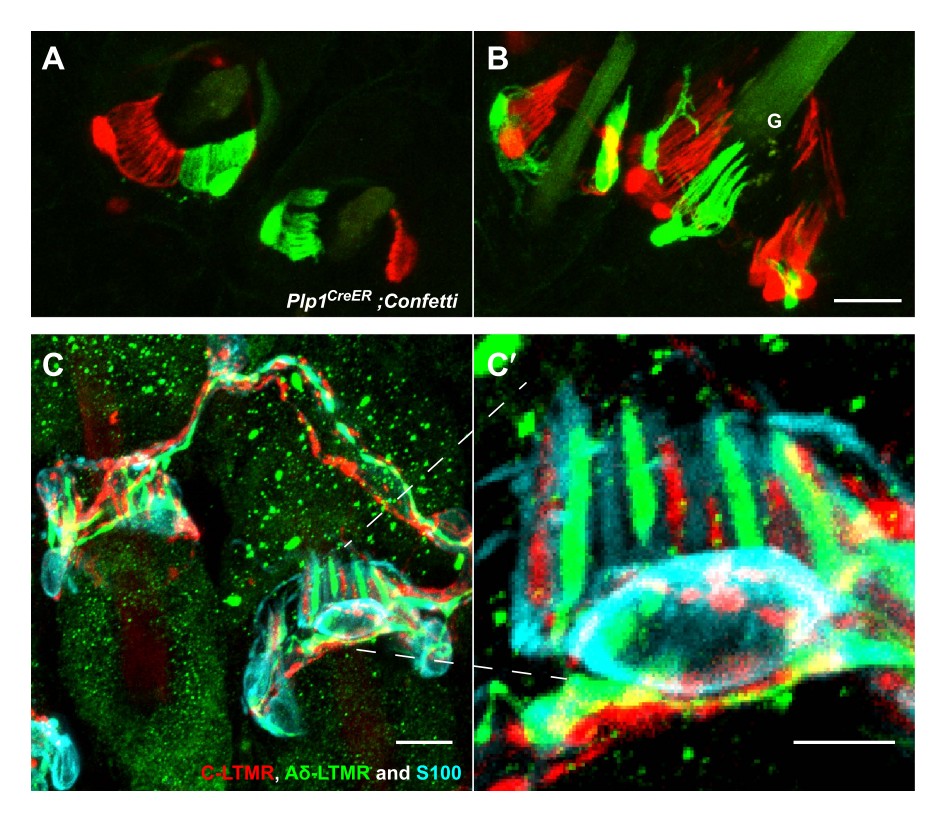

**Figure 3**. TSCs are tiled, and a single TSC can host axonal endings from multiple, physiologically distinct LTMR subtypes. (**A** and **B**) On back hairy skin sections taken from *Plp1^CreER*;*Confetti* animals, TSCs are randomly labeled with either green or red fluorescence. Neighboring TSCs at both guard (labeled with 'G' in **B**) and non-guard (**A**) hair follicles are tiled, exhibiting no overlapping processes. This experiment was done using 109 non-guard hair follicles and 23 guard hair follicles in two animals, all of which exhibited mosaic fluorescence labeling. 100% of these hair follicles exhibited tiled arrangements of TSCs at individual hair follicles. (**C** and **C'**) On back hairy skin sections from *Th^CreER*;*Rosa26^tdTomato*;*TrkB^tauEGFP* mice, C-LTMRs were labeled with tdTomato fluorescence (red), Aδ-LTMRs were labeled with anti-GFP (green) and TSCs were stained with anti-S100 (cyan). Shown here is an example in which C-LTMR and Aδ-LTMR endings associate with different processes of the same TSC at a zigzag or awl/auchene hair follicle. Thus, a single TSC hosts more than one type of LTMR axonal terminal. Four mice were used for the triple labeling experiment and identical results were observed in each. **C'** shows higher magnification of the TSC in the middle of the lanceolate complex shown in **C**. Scale bars, 20 µm for **A** and **B**, 10 µm in **C**, 5 µm in **C'**. Animals around 3 weeks of age were used for these experiments.

The following figure supplements are available for figure 3:

**Figure supplement 1**. A single TSC hosts axonal endings from Aβ, RA-LTMRs, and C-LTMRs.

from *Th^CreER*;*Rosa26^tdTomato*;*Npy2r-GFP* mice in which C-LTMRs and Aβ RA-LTMRs were labeled with tdTomato and GFP, respectively (**Figure 3—figure supplement 1**). Thus, TSCs are tiled, and the processes of an individual TSC encase the axonal endings of multiple, physiologically distinct LTMR subtypes.

## Ultrastructural features of lanceolate complexes at guard, awl/auchene, and zigzag hair follicles

We next investigated the ultrastructural basis of mechanical responsiveness of lanceolate endings, and in particular the ultrastructural relationships between Aβ RA-LTMR, Aδ-LTMR, and C-LTMR axonal terminals, TSCs, and the three hair follicles types. Transverse, ultrathin sections of lanceolate complexes associated with guard, awl/auchene, and zigzag hair follicles were collected, stained, and examined by electron microscopy (EM). Lanceolate complexes at all three hair follicle types appear as arrays of

individual units composed of blade-shaped TSC processes and LTMR axonal endings aligned against the basal lamina of the hair follicle (*Figure 4A*). At guard hair follicles, each Aβ RA-LTMR lanceolate axonal terminal (pseudo-colored in green) is encased by TSC processes (colored in pink) on three sides (*Figure 4B*). Thus, cross sections of Aβ RA-LTMR lanceolate complexes appear triangular in shape. There are frequent gaps between the TSC processes at the three edges of the axon. Interestingly, at the gaps or openings facing the hair follicle, the Aβ RA-LTMR axon terminals form small protrusions that are closely aligned, together with adjacent TSC processes, against the basal lamina of the hair follicle epithelial (*Figure 4B,B′*, arrows). In comparison, axonal protrusions through the other gaps, on sides facing away from the hair follicle epithelial cell, are less pronounced or absent and, when present, variable in width. Furthermore, the cytoplasm of Aβ RA-LTMR axons is densely packed with mitochondria. At awl/auchene and zigzag hair follicles, axonal terminals are encased on two sides by TSC processes, and thus cross sections of these lanceolate complexes appear spindle-shaped (*Figure 4C,D*). Compared to lanceolate complexes at guard hair follicles, TSC processes associated with awl/auchene and zigzag follicles tend to be thicker (widths of EM cross sections of TSC processes at guard hairs: 302.7 ± 132.7 nm, 17 process sections from two mice; awl/auchene hairs: 562.3 ± 76.5 nm, 21 process sections from two mice; zigzag hairs: 397.7 ± 19.4 nm, 32 process sections from three mice; these comparisons are not statistically significant), and the axonal processes they encase are thinner (areas of EM cross sections of axon terminals at guard hairs: $2.33 ± 0.06\ \mu m^2$, 5 axon sections from two mice; awl/auchene hairs: $0.76 ± 0.05\ \mu m^2$, 10 axon sections from two mice, p=0.002 compared with guard hairs; zigzag hairs: $0.57 ± 0.10\ \mu m^2$, 19 axon sections from three mice, p=0.001 compared with guard hairs). In addition, the gaps between TSC processes that expose the axonal endings on the side of hair follicle epithelial cells are larger than those found at guard hairs (*Figure 4C,C′,D,D′*, arrows) (widths of gaps of TSC processes toward guard hair follicles: 82.8 ± 13.9 nm, 7 axon sections from two mice; awl/auchene hair follicles: 162.9 ± 11.9 nm, 22 axon sections from two mice, p=0.05 compared with guard hairs; zigzag hair follicles: 196.6 ± 11.9 nm, 35 axon sections from three mice, p=0.01 compared with guard hairs). Interestingly, at awl/auchene and zigzag follicles, adjacent LTMR axonal endings often share the same TSC process; this was rarely observed for Aβ RA-LTMR endings at guard hairs. *Figure 4C* shows one such example in which three adjacent axon terminals associated with an awl/auchene hair follicle intervene among four TSC processes and appear bound together within a single unit. *Figure 4D* shows another example at a zigzag hair follicle, in which two adjacent axon terminals are tightly associated with three TSC processes to form a single unit (*Figure 4D*, lower left corner). An additional, distinguishing feature of lanceolate complexes at awl/auchene and zigzag hair follicles, compared to those at guard hairs, is that the ultrastructural properties of the lanceolate axons are heterogeneous. Indeed, some axons contain many mitochondria, similar to the Aβ RA-LTMR endings at guard hair follicles, whereas others contain few mitochondria (*Figure 4C,D*; *Figure 6—figure supplement 2A*). This ultratructural feature of axon terminals is observed across more than 30 serial EM sections, which spans more than 3 μm along the longitudinal axis of a lanceolate complex (*Figure 5A,B*). Since lanceolate complexes at awl/auchene hair follicles are triply innervated by Aβ RA-, Aδ-, and C-LTMRs while those at zigzag hair follicles are innervated by Aδ- and C-LTMRs, axons exhibiting different abundances of mitochondria may derive from distinct LTMR subtypes. Thus, the cellular and ultrastructural properties of lanceolate complexes differ among the hair follicle subtypes, and these differences may underlie unique sensitivities or responses of each LTMR subtype following skin indentation or hair deflection.

We next sought to compare the morphological and ultrastructural properties of individually defined Aβ RA-LTMR, Aδ-LTMR, and C-LTMR endings using EM. We found that *Wnt1-Cre;TrkA^{f/f}* mice, in which the nerve growth factor (NGF) receptor gene *TrkA (Nrtk1)* is selectively ablated in DRG neurons and other cells of neural crest origin, display a complete loss of C-LTMRs (*Figure 6A,A′*). Unexpectedly, lanceolate endings of Aβ RA-LTMRs at awl/auchene and guard hair follicles are also missing in *Wnt1-Cre;TrkA^{f/f}* mice (*Figure 6B,B′*), although Aβ RA-LTMR neurons labeled by Npy2r-GFP are still present in DRGs (*Figure 6—figure supplement 1*). Thus, C-LTMR and Aβ RA-LTMR endings in the skin are dependent on NGF–TrkA signaling for development. In contrast, the axonal endings of Aδ-LTMRs at both awl/auchene and zigzag hair follicles remain intact in these mutant mice (*Figure 6C,C′*). Therefore, by visualizing lanceolate complexes at zigzag hairs by EM in control and *Wnt1-Cre;TrkA^{f/f}* animals, we can distinguish between the C-LTMR endings that are missing in *Wnt1-Cre;TrkA^{f/f}* mutants, and Aδ-LTMRs endings that are intact. Interestingly, while axons associated with zigzag hair follicles in control mice have variable numbers of mitochondria, nearly all of the axons at zigzag hair follicles in *Wnt1-Cre;TrkA^{f/f}*

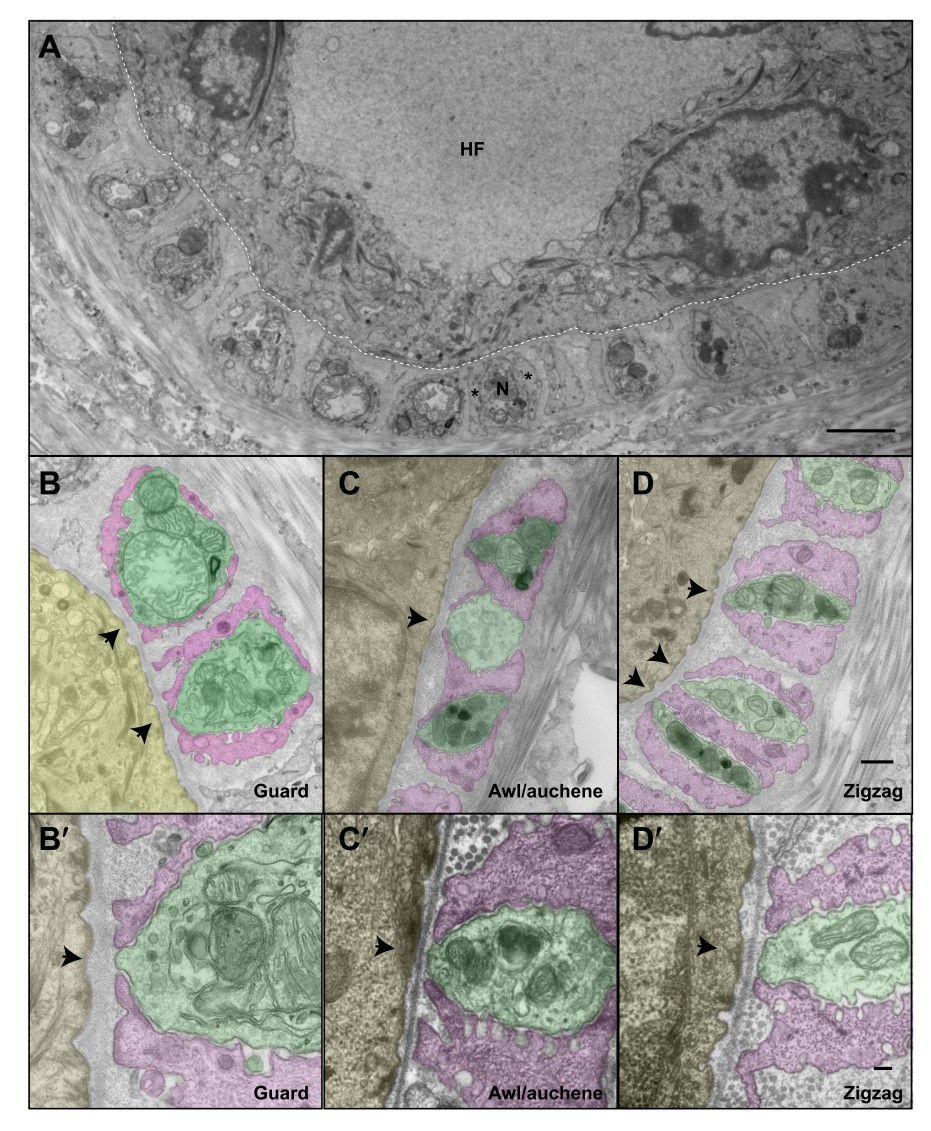

**Figure 4**. The ultrastructural relationships between LTMRs, TSCs, and hair follicle epithelial cells at the three hair follicle subtypes. (**A**) A transmission electron microscopic image of a cross section through a lanceolate complex at a guard hair follicle. Repeating units of axon terminals and TSC processes are regularly arranged in a single layer surrounding the hair follicle (HF). In each unit, a mitochondria-rich axonal terminal (N) is encased by TSC processes (*). (**B**) A cross section of the same guard hair follicle shown in **A**. Axon terminals are pseudo-colored in green; TSC processes are colored in pink; the hair follicle epithelial cell is colored in yellow. Each unit is composed of one axonal terminal encased by two or three TSC processes on three sides. Axon terminals contain a large number of mitochondria. Note that small protrusions of axons (arrows) and TSC processes are precisely aligned against the basal lamina of the hair follicle. (**C**) A cross section of a lanceolate complex of an awl/auchene hair follicle. Each axon terminal is encased by two TSC processes on two sides. More than one axonal ending and its associated TSC processes is often packed into a single 'unit'. Shown here is one unit composed of three axon terminals intervening among four TSC processes. Compared to guard hair follicles, awl/auchene hair follicles exhibit more area of exposed axon terminal membrane facing the outer root sheath cell of the follicle (arrow). In addition, TSC processes are thicker than those at guard hair follicles, while axon diameters are smaller. Also, the axon terminals appear heterogeneous: some have few mitochondria (the middle axon) and some many mito-chondria (the other two axons). (**D**) A cross section of a lanceolate complex associated with a zigzag hair follicle. Similar to awl/auchene hair follicles, each axon terminal is encased by two TSC processes on two sides. One of the units in the lower left corner is composed of two axon terminals encased by three TSC processes. As with the awl/auchene complexes, and in comparison to guard hairs, areas of exposed axon terminal membrane adjacent

*Figure 4. Continued on next page*

*Figure 4. Continued*

to the hair follicle epithelial cell are large (arrows). In addition, similar to awl/auchenes, the zigzag TSC processes are thicker and axonal sections are smaller compared to guard hair follicles. Axon terminals also display varying amounts of mitochondria. (**B′**, **C′** and **D′**) High magnification EM images of cross sections of guard (**B′**), awl/auchene (**C′**), and zigzag (**D′**) hair follicles show the gaps between TSC processes and the axon protrusions in between (arrows). Animals around 4 weeks of age were used for these experiments. Scale bars, 2 μm in **A**; 500 nm for **B**, **C** and **D**; 100 nm for **B′**, **C′** and **D′**.

mice exhibit a high density of mitochondria (*Figure 6D,E*; *Figure 6—figure supplement 2B*). Assuming that the *TrkA* mutation has no effect on mitochondria density in Aδ-LTMR axonal terminals, this finding indicates that the lanceolate endings with few mitochondria belong to C-LTMRs while those with abundant mitochondria belong to Aδ-LTMRs or Aβ RA-LTMRs (*Figure 5*; *Figure 6F–H*).

Several additional interesting ultrastructural features of LTMR lanceolate complexes at all three hair follicle subtypes were observed. In particular, numerous small vesicles are located within the cytoplasm of lanceolate endings of all LTMR subtypes (*Figure 7A–C*, arrows). Recent studies suggest that these are glutamate-containing vesicles, which may mediate communication with TSCs and thereby regulate lanceolate complex assembly and maintenance (*Woo et al., 2012*; *Banks et al., 2013*). Additionally, many pinocytotic vesicles are associated with both the inner and outer surfaces of TSC processes (*Figure 7A–C*, arrowheads). Similar observations have been made for lanceolate complexes of hair follicles from facial skin or ear hairy skin in mouse, rat, and humans (*Yamamoto, 1966*; *Cauna, 1969*; *Hashimoto, 1972*; *Kaidoh and Inoue, 2000*), although the sensory neuron subtype under investigation in those studies was not known. Additionally, a network of longitudinally-oriented collagen fibers can be observed in the extracellular space between lanceolate endings and basement membrane of the follicle and appear as circular structures in EM cross-sections (*Figure 6D*, arrow). Another network of horizontally oriented collagen fibers appear as stripes around longitudinal lanceolate endings in the outer circle, presumably surrounding circumferential endings (*Figure 6D*, arrowhead) (*Low, 1962*; *Parakkal, 1969*). Such an alignment of collagen fiber networks may provide structural rigidity important for transferring hair follicle deflection into distortion or compression of lanceolate complexes and thereby activation of LTMRs.

To further assess the ultrastructural relationship between physiologically distinct lanceolate complex subtypes and the three hair follicle types, we prepared hair follicle specimens for EM analysis using tannic acid treatment, which preserves morphology and enhances visualization of extracellular matrix components (*Dingemans and van den Bergh Weerman, 1990*). Intriguingly, there is an abundance of prominent hemidesmosomes located along the outer membranes of the outer root sheath epithelial cells of the three hair follicle subtypes (*Figure 7D–G*, white arrow heads). Fine filament-like structures emanate from these hemidesmosomes, pass through the basal lamina, and appear to form direct contacts with the membranes of Aβ RA-LTMR, Aδ-LTMR, and C-LTMR lanceolate axonal endings as well as TSC processes. These tether-like filaments are approximately 100 nm in length and may be similar to anchoring filaments or anchoring fibrils, which are essential components of dermo–epidermal junctions connecting basal keratinocytes with dermal cell types (*Keene et al., 1987*; *Regauer et al., 1990*; *Burgeson and Christiano, 1997*). It is tempting to speculate that these filamentous connections between hair follicle epithelial cells and both LTMR lanceolate endings and TSCs are involved in assembly, maintenance, or mechanotransduction at lanceolate complexes. Consistent with the latter, 100-nm long, tether-like structures were observed in DRG sensory neurons grown in cell culture, where they may be required for mechanosensitive currents of DRG sensory neurons (*Hu et al., 2010*).

## The roles of LTMRs and TSCs in the maintenance of lanceolate complex integrity

The intricate structural relationship between LTMRs and TSCs suggests that both cells types are essential for function and maintenance of lanceolate complexes at guard, awl/auchene, and zigzag hair follicles. However, the relative contributions of LTMRs and TSCs for the maintenance of lanceolate complex integrity are unknown. To address this, we first examined the integrity of LTMR axonal terminals following genetic ablation of Schwann cells. We crossed the glial-specific *Plp1^CreER* mouse line to the *Rosa26^hDTR* reporter line, which expresses the human diphtheria toxin receptor (hDTR) in a Cre-dependent manner (*Buch et al., 2005*). *Plp1^CreER;Rosa26^hDTR* animals were treated with tamoxifen at

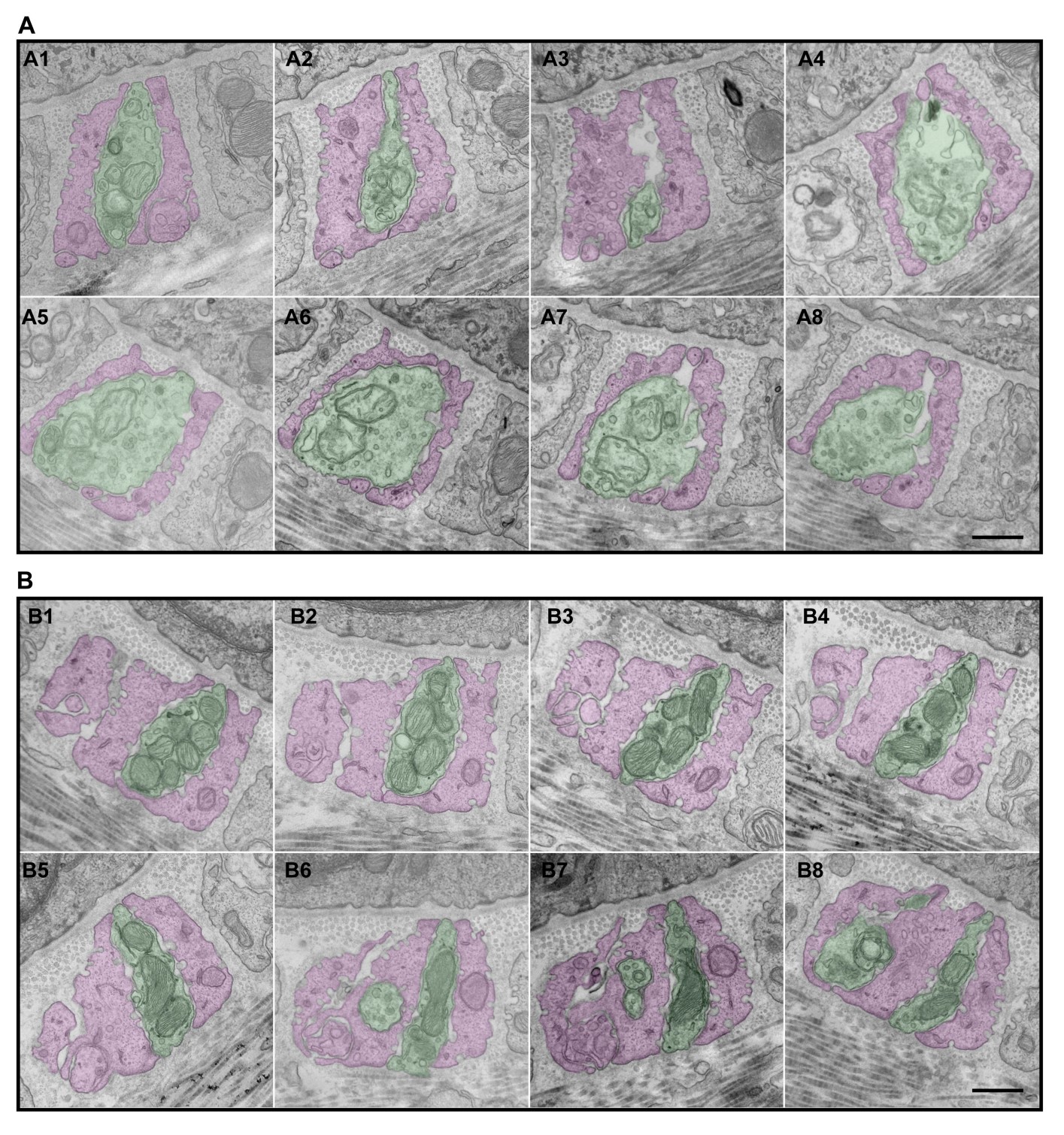

**Figure 5**. Serial EM cross-sections of C-LTMR and Aδ-LTMR axonal endings at a zigzag hair follicle. Serial EM cross-sections spanning more than 3 μm along two different LTMR axonal terminals of lanceolate complex at a zigzag hair follicle were collected. Representative images that are approximately 0.4 μm to 0.5 μm apart were shown. (**A**) A1 to A8 are serial EM cross-sections of an axonal terminal that has a relatively small number of mitochondria with low electron density. Further analyses (**Figure 6**, **Figure 6—figure supplement 2**) show that this axonal ending type is likely to be C-LTMR. (**B**) B1 to B8 are serial EM cross-sections of an axonal terminal that is packed with mitochondria with relatively high electron density. Further analyses (**Figure 6**, **Figure 6—figure supplement 2**) show that this axonal ending type is likely to be Aδ-LTMR. Animals around 4 weeks of age were used for these experiments. Scale bar, 500 nm.

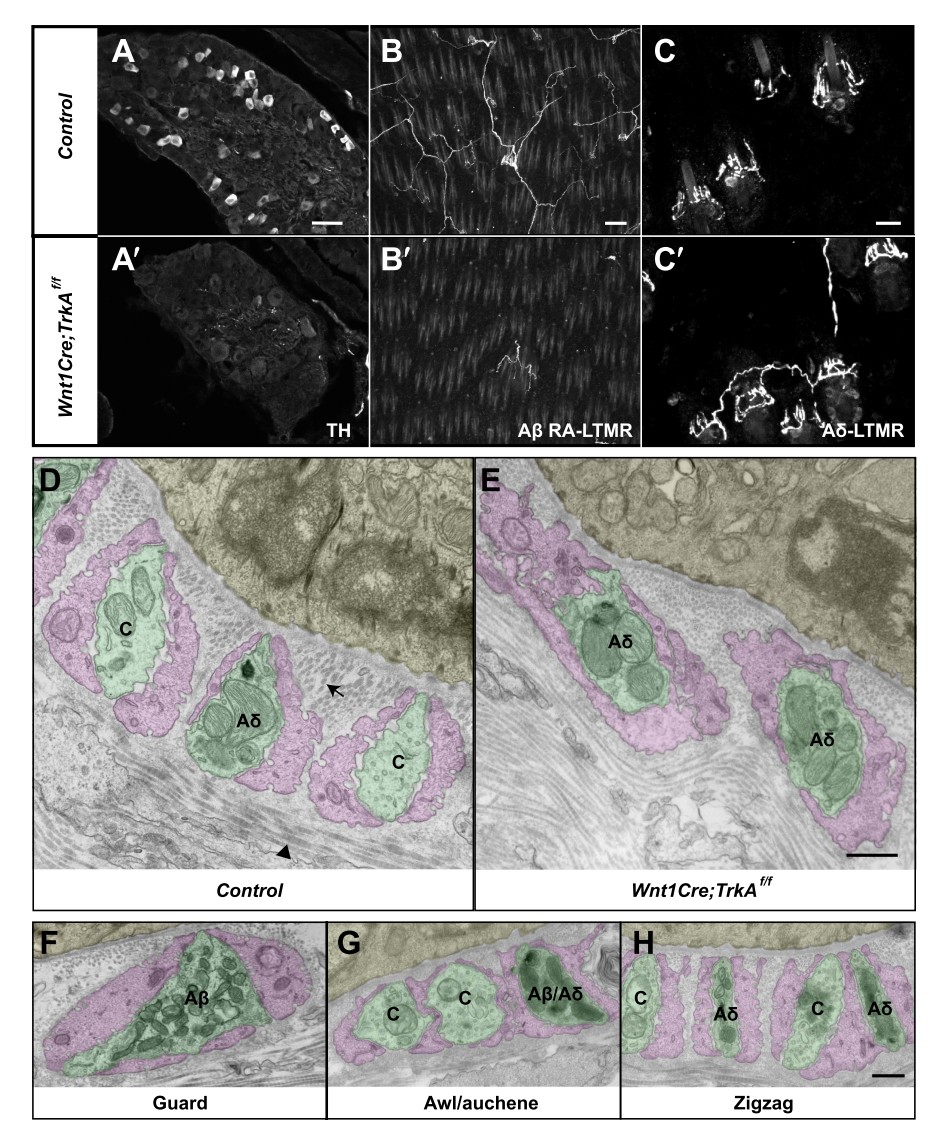

**Figure 6**. Identification of C-LTMR, Aδ-LTMR, and Aβ RA-LTMR axonal endings using EM. (**A** and **A'**) DRG sections from *Wnt1Cre;TrkA^f/f* (**A'**) and control (**A**) animals were stained with anti-TH. TH+ C-LTMRs are nearly completely lost in TrkA conditional knockout animals compared to control. (**B** and **B'**) Whole-mount GFP immunostaining of back skin samples from *Wnt1Cre;TrkA^f/f;Npy2r-GFP* (**B'**) and control (**B**) animals shows that cutaneous innervation of Npy2r-GFP+ Aβ RA-LTMRs at hair follicles is almost completely lost in TrkA conditional knockout animals compared to control. (**C** and **C'**) GFP immunostaining of back skin sections from *Wnt1Cre;TrkA^f/f;TrkB^tauEGFP* (**C'**) and *TrkB^tauEGFP* control (**C**) animals shows that innervation of hair follicles by TrkB^tauEGFP+ Aδ-LTMRs remains intact in the TrkA conditional knockout animals compared to control. (**D** and **E**) Cross sections of lanceolate complexes at zigzag hair follicles from *Wnt1Cre;TrkA^f/f* (**E**) and control (**D**) mice. Axon terminals are pseudo-colored in green; TSC processes are colored in pink; hair follicle epithelial cells are colored in yellow. As shown in ***Figure 4D***, axon terminals at the control zigzag hair follicle have varying numbers of mitochondria. In contrast, all axons at the zigzag hair follicle from *Wnt1Cre;TrkA^f/f* animals exhibit abundant clusters of mitochondria. Thus, axons containing few mitochondria are C-LTMRs, which are lost in TrkA conditional knockout animals, whereas axons containing abundant mitochondria are Aδ-LTMRs, which remain intact in TrkA conditional knockout animals. (**F**) A cross section of a lanceolate complex at a wild-type guard hair follicle. All axons associated with guard hair lanceolate complexes are densely packed with mitochondria and are derived from Aβ RA-LTMRs. (**G**) A cross section of a lanceolate complex at a wild-type awl/auchene hair follicle. Axons with few mitochondria are C-LTMRs; axons with abundant mitochondria are either Aβ RA-LTMRs or Aδ-LTMRs. (**H**) A cross section of a lanceolate complex at a wild-type zigzag hair follicle.
*Figure 6. Continued on next page*

*Figure 6. Continued*

Axons with few mitochondria are C-LTMRs, whereas axons with abundant mitochondria are Aδ-LTMRs. Animals around 4 weeks of age were used for these experiments. Scale bars, 50 µm for **A** and **A'**; 100 µm for **B** and **B'**; 20 µm for **C** and **C'**; 500 nm for **D–H**.

The following figure supplements are available for figure 6:

**Figure supplement 1**. Aβ RA-LTMR neurons remain intact in *Wnt1Cre;TrkA^{f/f}* animals.

**Figure supplement 2**. Quantification of mitochondrial abundance.

P16 to activate Cre recombinase and thus glial cell-specific expression of hDTR. These animals were then subjected to injections of DTX at P24 and P27, which results in selective ablation of hDTR-expressing cells (*Saito et al., 2001*; *Cavanaugh et al., 2009*). We observed efficient ablation of TSCs at hair follicles in DTX-treated *Plp1^{CreER};Rosa26^{hDTR}* animals, compared to DTX-treated control animals (*Figure 8*). In these experiments, TSCs were visualized using tdTomato fluorescence in animals harboring the *Rosa26^{tdTomato}* allele (*Figure 8*). Remarkably, 100% of the hair follicles lacking TSCs exhibited a complete loss of longitudinal lanceolate endings and a partial loss of circumferential endings labeled by Tuj1 and NFH immunostaining (*Figure 8A,A',B,B'*). A partial loss of CGRP+ circumferential endings was also observed (*Figure 8C,C'*). These findings indicate that LTMR axon terminals retract from hair follicles following ablation of TSCs. The disappearance of longitudinal lanceolate endings is coincident with degeneration of TSCs, while circumferential lanceolate endings are more resilient and remain in the absence of TSCs (*Figure 8—figure supplement 1*). The partial loss of CGRP+ circumferential axons in the absence of TSCs may be due to instability of sensory–neural complexes at hair follicles that results from the loss of TSCs and LTMR lanceolate endings. The PLP gene is expressed in almost all glial subtypes in the periphery (*Jessen and Mirsky, 2005*). Due to limitations of our genetic tools, we cannot further assess the contributions of other glial cell types, such as myelinating and non-myelinating Schwann cells to the anchoring of LTMR endings at hair follicles. However, the fact that LTMR longitudinal lanceolate endings at hair follicles are severely affected regardless of whether they are heavily myelinated (Aβ RA-LTMR), lightly myelinated (Aδ-LTMR) or unmyelinated (C-LTMR), whereas unmyelinated free nerve endings remain relatively intact (*Figure 8A,A'*, white squares), strongly suggests that TSCs play a major role in maintaining the integrity of lanceolate complexes at hair follicles.

We next asked the reciprocal question; are lanceolate axonal endings required for the maintenance of TSCs? To denervate hair follicles in back hairy skin, dorsal cutaneous nerves that innervate the right side of the back of adult animals were severed, while nerves innervating the left side remained intact, providing a control. As expected, 17 days following surgery, cutaneous nerves innervating the right, surgically denervated side of the animal had completely degenerated (*Figure 9A'*), whereas cutaneous innervation of the left, control side was unaffected (*Figure 9A*). Strikingly, TSCs, labeled by S100 immunostaining, remained intact and in close association with hair follicles in the absence of LTMR endings (*Figure 9B,B'*). Even 44 days following skin denervation, TSCs at denervated hair follicles remained intact (*Figure 9C,C',D,D',E,E'*). In contrast, both Schwann cells associated with circumferential axonal terminals at hair follicles (*Figure 9D,D'*; white arrow) and myelinating Schwann cells in the dermis, visualized by S100 and MBP immunostaining (*Figure 9E,E'*), had degenerated in the absence of cutaneous nerves. Similar observations were made when TSCs were genetically labeled using *Plp1^{CreER};Rosa26^{GCaMP3}* mice that had received tamoxifen treatment 'prior' to surgical denervation. As in the aforementioned S100 immunostaining experiments, GFP immunostaining of skin sections (to detect GCaMP3) showed that TSCs that were genetically labeled prior to denervation are morphologically intact following surgical denervation of the skin (*Figure 9F,F'*). In addition, GFP+ Schwann cells that did not express either S100 or MBP were also observed in the dermis, indicating the presence of PLP-expressing immature Schwann cells that may function to regenerate myelin sheaths (*Figure 9F'*, *Figure 9—figure supplement 1*). We quantified the number of TSCs at individual guard hair and non-guard hair follicles (awl/auchene and zigzag hairs combined) 5 weeks following skin denervation. This quantitative assessment revealed that, despite complete loss of cutaneous LTMR endings on the denervated side of the animal, comparable numbers of TSCs were present at hair follicles on control and

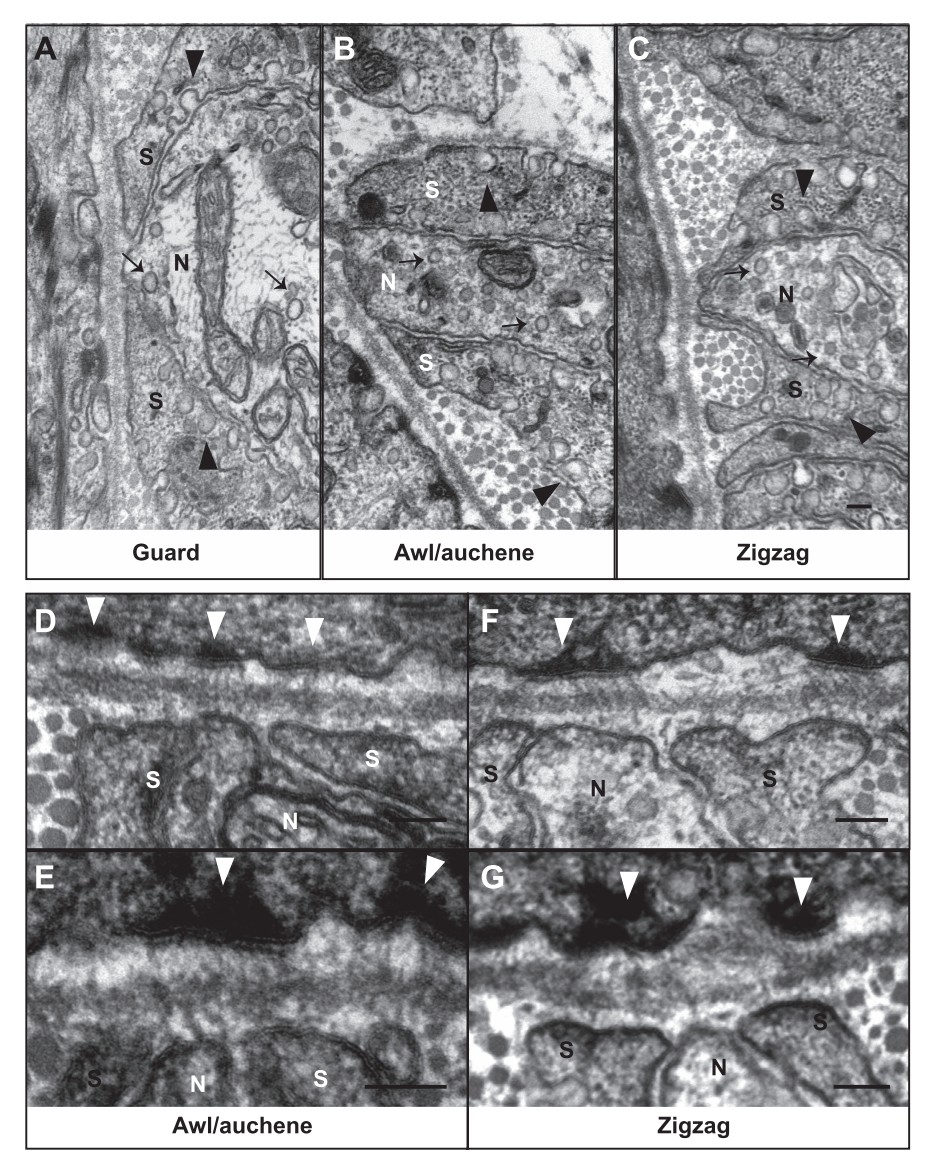

**Figure 7**. Ultrastructural features of lanceolate complexes revealed by EM using tannic acid-treated specimens. (**A–C**) Cross sections of lanceolate complexes at guard, awl/auchene, and zigzag hair follicles. Small vesicles can be observed within axon terminals (arrows in **A–C**). TSC processes contain fine filaments that are nearly parallel to the long axis of the follicle and therefore appear as dark spots within the cytoplasm. Numerous pinocytotic vesicles are associated with both the inner and outer surfaces of TSC processes (arrowheads in **A–C**). (**D–G**) Hemidesmosomes are seen along plasma membranes of hair follicle outer root sheath cells that face LTMR axons and TSC processes (white arrowheads). Fine filament-like structures emanate from the hemidesmosomes, traverse the basal lamina, and form contacts with LTMR axon terminals and TSC processes. Axon terminals are labeled with 'N'; TSC processes are labeled with 'S'. Animals around 4 weeks of age were used in these experiments. Scale bars, 100 nm.

denervated sides (*Figure 9G*). Therefore, TSCs associated with Aβ RA-LTMRs, Aδ-LTMRs, and C-LTMRs, maintain both their location and structural integrity in the absence of axonal processes in adult mice, for at least several weeks.

A few weeks following skin denervation, re-innervation of some hair follicles was observed. In contrast to 7 days after denervation, where few fibers were seen in the denervated skin, by 33 days after denervation, approximately 70% of the denervated skin had been innervated (*Figure 10A,B*). The majority of re-innervation most likely derives from axons that had regenerated from the severed dorsal

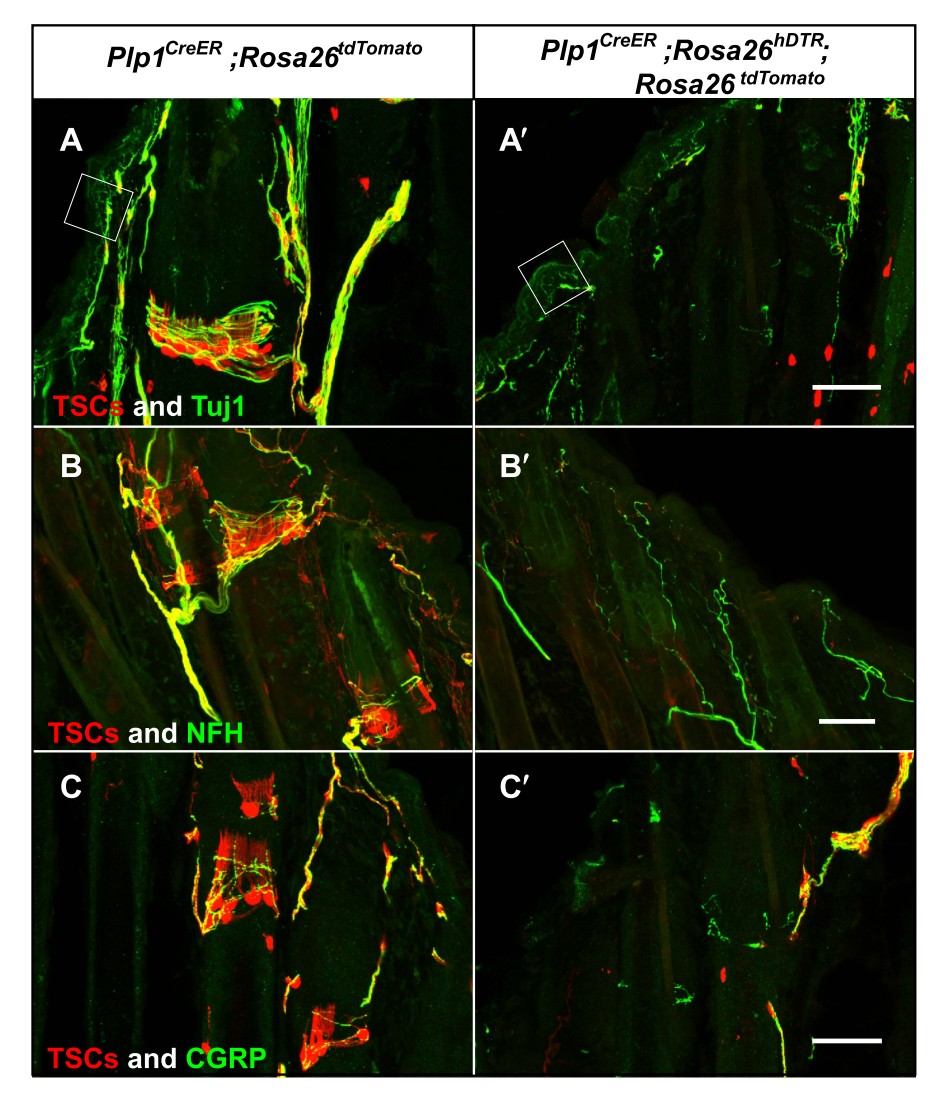

**Figure 8**. Genetic ablation of TSCs leads to loss of LTMR innervation at hair follicles. In skin sections from *Plp1^CreER^;Rosa26^tdTomato^* (**A**–**C**) and *Plp1^CreER^;Rosa26^hDTR^;Rosa26^tdtomato^* (**A'**–**C'**) mice, TSCs were visualized by tdTomato fluorescence. In *Plp1^CreER^;Rosa26^hDTR^;Rosa26^tdtomato^* animals, treatments with tamoxifen and DTX lead to complete loss of TSCs at hair follicles (**A'**–**C'**). In the absence of TSCs, Tuj1 (**A** and **A'**, green) and NFH staining (**B** and **B'**, green) shows a complete loss of longitudinal axonal terminals and a partial loss of circumferential axons at the presumptive region for lanceolate complexes; CGRP$^+$ peptidergic nociceptor fibers were also partially lost (**C** and **C'**, green). Small white squares at **A** and **A'** highlight free nerve endings in the epidermis and dermis. Animals around 4 weeks of age were used in these experiments. Scale bars, 50 μm.

The following figure supplements are available for figure 8:

**Figure supplement 1**. Loss of LTMR endings is coincident with degeneration of TSCs.

cutaneous nerves (*Jackson and Diamond, 1984*). For the skin regions close to the edges of the denervated territory, the re-innervating fibers may also derive from adjacent intact nerves. Re-innervation in the center of the denervated skin was least robust, which may due to the long distance between that skin region and the cut sites of T9 and T10 dorsal cutaneous nerves (*Figure 10B*). Nevertheless, re-innervation of most skin regions was prominent by 33 days after denervation (*Figure 10B*). This finding of re-innervation of denervated hair follicles, together with the observation that TSCs are remarkably stable following skin denervation, prompted us to ask whether TSCs that are associated

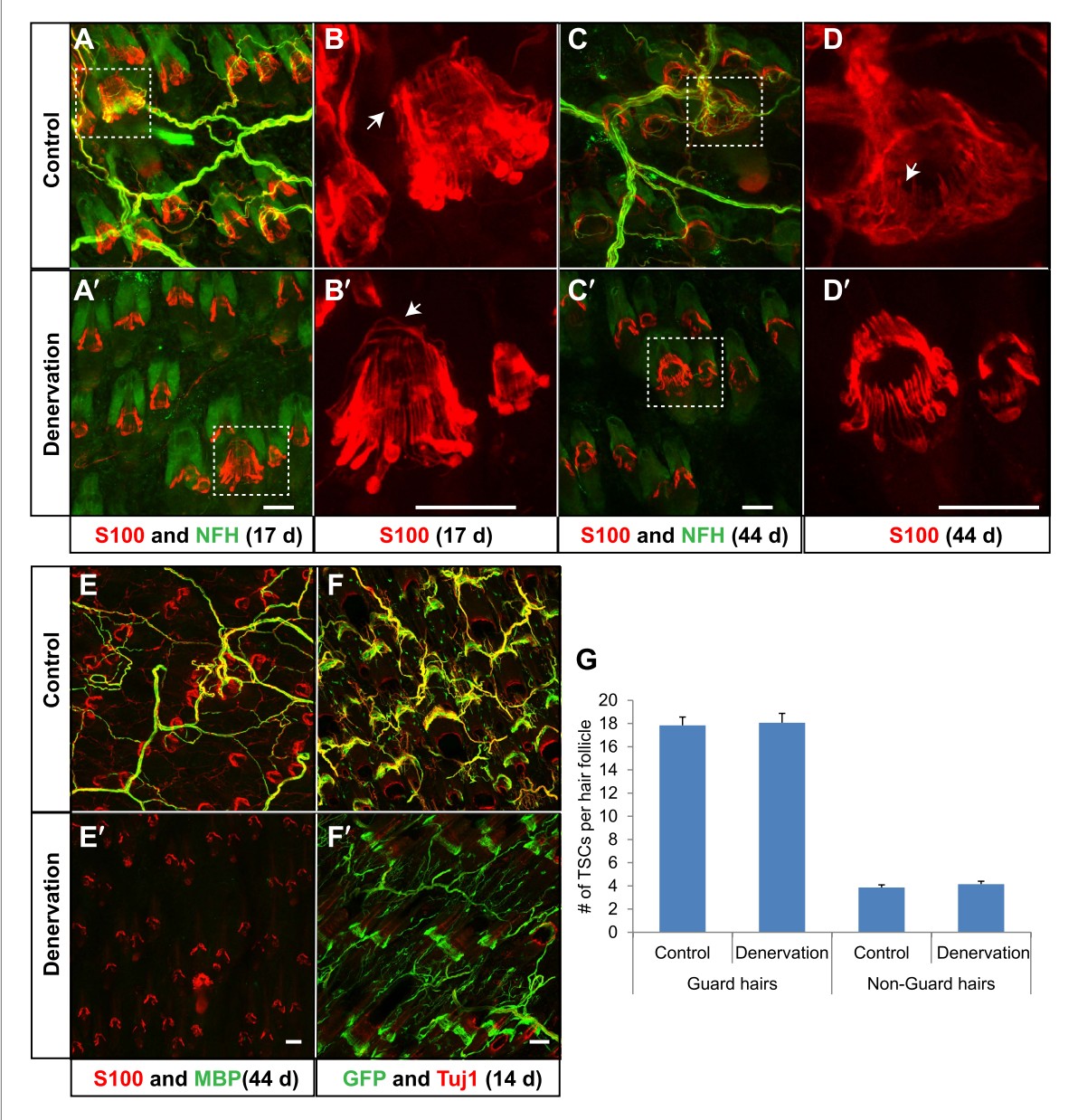

**Figure 9**. TSCs remain intact and associated with hair follicles following dorsal cutaneous nerve axotomy and distal axon degeneration. (**A** and **A'**) Whole-mount immunostaining of S100 (red) and NFH (green) shows that 17 days after dorsal cutaneous nerve axotomy, while NFH+ cutaneous nerve have completely degenerated in the denervated skin (**A'**, right back skin) compared to the control skin (**A**, left back skin), TSCs remain intact and associated with hair follicles (**A'**). (**B** and **B'**) Enlarged views of boxed regions in **A** and **A'**. At control hair follicles (**B**), TSCs are associated with longitudinal lanceolate endings and circumferential endings (arrow). At denervated hair follicles (**B'**), TSCs that were associated with longitudinal lanceolate endings prior to denervation appear normal, while TSCs associated with circumferential axonal terminals (arrows) are partially lost (arrow). (**C** and **C'**) Whole-mount immunostaining of S100 (red) and NFH (green) shows that, 44 days after axotomy, TSCs in the denervated skin (**C'**) remain intact. (**D** and **D'**) Enlarged views of boxed regions in **C** and **C'** show that compared to TSCs in the control skin (**D**), TSCs that were associated with longitudinal lanceolate endings at denervated hair follicles (**D'**) appear normal, while TSCs associated with circumferential axonal terminals are completely lost (arrow in **D**). (**E** and **E'**) Whole-mount immunostaining of S100 (red) and MBP (green) shows that, at 44 days after axotomy, while myelinating Schwann cells have completely degenerated, TSCs in the denervated skin remain intact (**E'**). (**F** and **F'**) In *Plp1^CreER;Rosa26^GCamp3* animals, TSCs were induced to express GCaMP3 by tamoxifen injection before dorsal cutaneous nerve axotomy. Immunostaining for GFP (green) and Tuj1 (red) shows that, at 14 days following axotomy, while Tuj1+ cuaneous axons have completely degenerated, genetically labeled TSCs remain intact (**F'**). (**G**) Quantification of TSC numbers 5 weeks after denervation surgery shows no changes in TSC numbers at both guard hair and non-guard hair follicles in denervated skin compared to

*Figure 9. Continued on next page*

*Figure 9. Continued*

those at the control skin. n = 3. Animals around 8 weeks of age were used for all whole-mount immunostaining experiments in **A**–**E'** and **G**. *Plp1^{CreER}*; *Rosa26^{GCamp3}* mice around 4 weeks old were used in experiments shown in **F** and **F'**. Scale bars, 50 μm.

The following figure supplements are available for figure 9:

**Figure supplement 1**. Genetically labeled immature Schwann cells after skin denervation.

with LTMR lanceolate complexes prior to denervation, and remain following denervation, become re-innervated by sprouting or regenerating LTMR axons. To test this possibility, TSCs were genetically labeled using *Plp1^{CreER}*;*Rosa26^{GCaMP3}* animals and tamoxifen application prior to denervation. TSCs and axonal processes in denervated back skin were then observed using GFP and Tuj1 double immunostaining at different time points after the surgery. In control hair follicles on the non-denervated side of the back skin, longitudinal lanceolate endings were associated with TSC processes and formed typical, intricate longitudinal lanceolate complexes at guard hair and non-guard hair follicles; circumferential endings were also observed surrounding these longitudinal lanceolate complexes at all hair follicles (*Figure 10C*). On the denervated side of the back skin, though most longitudinal lanceolate endings were still absent three weeks following skin denervation, prominent re-innervation of guard hair follicles in the form of thick circumferential endings and occasional longitudinal lanceolate endings is readily observed (*Figure 10D*, arrow). A lesser amount of re-innervation in the form of circumferential endings was also seen at this time point in surrounding non-guard hair follicles (*Figure 10D*). 1 week later, at week four following skin denervation, prominent circumferential and longitudinal axonal endings were readily observed at the re-innervated guard hair follicles (*Figure 10E*). Circumferential endings were also observed in the surrounding non-guard hair follicles at this time point. At 5 weeks following denervation, sensory endings associated with re-innervated guard hair follicles were morphologically comparable to those in control skin (*Figure 10F*). Overall, approximately 90% of hair follicles exhibited at least some re-innervation by 5 weeks after denervation. Lanceolate complexes at guard hairs have considerably more advanced recovery compared to non-guard hair follicles. Most non-guard hair follicles, especially zigzag hair follicles, display only sparse circumferential endings at this stage (*Figure 10F*). We also addressed the specificity of axonal re-innervation of TSC complexes. For this, skin denervation experiments were done using *Npy2r-GFP* mice, in which Aβ RA-LTMRs are labeled with GFP (*Li et al., 2011*). In control skin, whole-mount immunostaining using anti-GFP confirmed previous findings that, as expected for Aβ RA-LTMRs (*Li et al., 2011*), Npy2r-GFP⁺ axon terminals form longitudinal lanceolate endings associated with guard and awl/auchene, but not zigzag hair follicles (*Figure 10G*, white arrowheads). Remarkably, 83 days following denervation, Npy2r-GFP⁺ axons had re-innervated awl/auchene and guard hair follicles in the denervated skin region (*Figure 10H*, white arrowheads). While some of the Aβ RA-LTMR axon terminals exhibited simple circumferential endings around hair follicles at this time point and excess sprouting of GFP⁺ axons innervating adjacent zigzag hair follicles was occasionally seen, the majority of GFP⁺ axons had developed typical longitudinal lanceolate endings associated with the denervated guard and awl/auchene hairs. In summary, lanceolate complex TSCs remain intact and tightly associated with hair follicles following skin denervation. Furthermore, these intact TSCs become re-associated with the lanceolate endings of regenerated LTMRs, and the specificity of the LTMR subtype/hair follicle relationship is largely if not completely preserved, at least for Aβ RA-LTMRs, following re-innervation. This observation raises the intriguing possibility that TSCs, which remain intact following nerve injury, serve to attract axonal endings of specific LTMR subtypes to restore proper structure and organization of Aβ RA-LTMRs, Aδ-LTMRs, and C-LTMRs lanceolate complexes at the three hair follicle types.

## Discussion

Hair follicle longitudinal lanceolate complexes, which are formed by Aβ RA-LTMR, Aδ-LTMR, and C-LTMR axonal endings and their associated TSCs, are the primary mechanically sensitive structures that transform hair follicle deflection into electrical impulses carried by the axonal branches of LTMR neurons to the CNS. In this study, we combined genetic labeling of LTMRs and TSCs, electron microscopic analyses, and surgical and genetic manipulations to characterize the organization, structure, maintenance, and regeneration of hair follicle lanceolate complexes of the mouse.

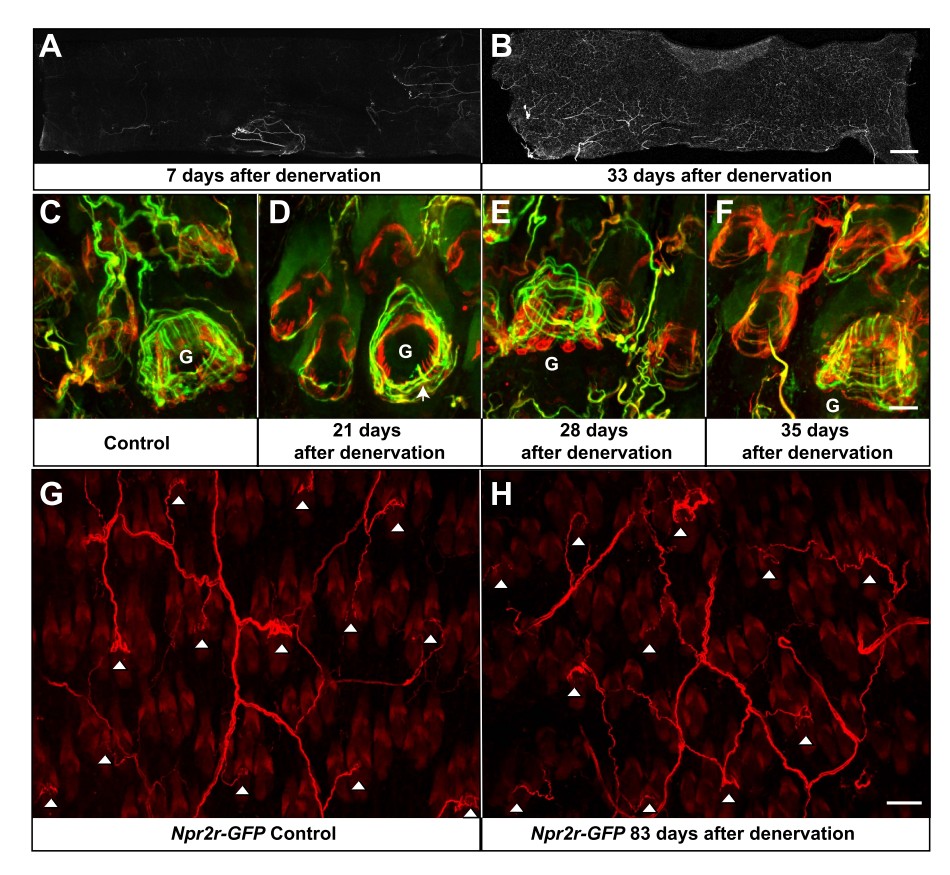

**Figure 10**. Re-innervation of TSCs following dorsal cutaneous nerve axotomy. (**A** and **B**) Whole-mount immunostaining with NFH shows that axons are missing from denervated skin 7 days after axotomy (**A**), whereas nerve fibers have begun extending into the denervated area 33 days after dorsal cutaneous nerve axotomy (**B**). (**C**) In the control skin of *Plp1^CreER^;Rosa26^GCamp3^* animals, whole-mount immunostaining with Tuj1 (green) and GFP (red) shows the typical morphology of lanceolate complexes at a guard hair follicle located in the lower right area of the image, as well as surrounding non-guard hair follicles. (**D**) 3 weeks after dorsal cutaneous nerve axotomy, re-innervation of guard hair follicles in the form of circumferential endings can be observed. A few longitudinal lanceolate endings can be occasionally observed (arrow). A small degree of re-innervation can also be seen in the surrounding non-guard hair follicles. (**E**) 4 weeks after dorsal cutaneous nerve axotomy, longitudinal lanceolate endings begin to form at re-innervated guard hair follicles. Circumferential endings can also be observed in the surrounding non-guard hair follicles. (**F**) 5 weeks after axotomy, lanceolate complexes at re-innervated guard hair follicles are comparable to those in the control, uninjured skin. More innervation of the surrounding non-guard hair follicles can also be seen. Guard hairs are labeled with 'G' in panels **C** to **F**. (**G** and **H**) *Npy2r-GFP* animals were subjected to dorsal cutaneous nerve axotomy, as above. **G** shows whole-mount GFP immunostaining of the control, non-denervated skin (left back hairy skin), while **H** shows GFP immunostaining of denervated skin (right back hairy skin) 83 days after axotomy. In control, uninjured skin, GFP+ Aβ RA-LTMRs form longitudinal lanceolate endings at guard and awl/auchene hair follicles (**G**, white arrowheads). 83 days after axotomy, GFP+ Aβ RA-LTMRs re-innervate guard and awl/auchene hair follicles in the denervated skin (**H**, white arrowheads), indicating that the specificity of the hair follicle subtype innervation pattern is conserved during the re-innervation processes. Many GFP+ axonal terminals in the re-innervated skin form longitudinal lanceolate endings that are comparable to those in the control skin. Scale bars, 2 mm for **A** and **B**; 20 μm from **C** to **F**; 50 μm for **G** and **H**.

## The cellular properties of hair follicle LTMR lanceolate complexes and implications for LTMR properties

Neurobiologists have long postulated that distinct morphological properties of mechanically sensitive end organs associated with LTMRs underlie the unique physiological properties of LTMR subtypes (*Johnson, 2001*; *Maricich et al., 2009*). Our findings modify this view and provide new insights into

the basis of LTMR subtype properties. Guard hair follicles have the longest hair shafts and are innervated by longitudinal lanceolate endings belonging exclusively to Aβ RA-LTMRs. If, as we believe, LTMR lanceolate endings and their associated TSC processes serve as detectors of hair follicle deflection, then guard hair follicles are armed with many more such detectors, displaying fewer but longer processes, compared to those associated with zigzag and awl/auchene hair follicles. Thus, the unique cellular and morphological properties of guard hair lanceolate complexes may endow guard hair-associated Aβ RA-LTMRs with unique functions or response properties during hair movement. On the other hand, our analysis of awl/achene and zigzag hair follicles indicates that cellular and morphological differences of longitudinal lanceolate complexes are unlikely to account for the unique physiological properties of Aβ RA-, Aδ-, and C-LTMRs at these hair follicle subtypes. Our mosaic genetic labeling of lanceolate complex TSCs shows that these cells are tiled at individual hair follicles and that the processes of adjacent TSCs do not overlap. Importantly, at both awl/auchene and zigzag hair follicles, a single TSC plays host to interdigitated axonal endings of two or more LTMR subtypes. Thus, while Aβ RA-, Aδ-, and C-LTMRs exhibit markedly different rates of adaptation to sustained stimuli, their axons form strikingly similar longitudinal lanceolate endings, they can innervate the same hair follicle and, as shown here for awl/auchene and zigzag hair follicles, they can even share the same TSC. These findings provide strong support for a model, in which the unique physiological properties and adaptation rates of Aβ RA-, Aδ-, and C-LTMRs associated with awl/auchene and zigzag hair follicles are due to intrinsic neuronal differences rather than unique end organ cell types or morphologies.

## The ultrastructural properties of hair follicle LTMR lanceolate complexes and implications for LTMR function and mechanosensation

The fine structural relationship between LTMR axonal endings, TSC processes, and hair follicle epithelial cells is likely a key determinant of LTMR mechanical sensitivity. Extensive light and electron microscopic analyses of lanceolate complexes were reported beginning in the 1960s (*Yamamoto, 1966*; *Cauna, 1969*; *Halata, 1993*). These previous studies, together with findings of the present study, indicate that the general morphology of lanceolate complexes associated with hair follicles from different body regions and between different species is highly conserved. In this study, for the first time, we report EM analysis that defines the ultrastructural properties of Aβ RA-LTMR, Aδ-LTMR, and C-LTMR lanceolate complexes and their associations with the three main hair follicle types; guard, awl/auchene, and zigzag hair follicles. Our analysis revealed that the ultrastructural properties of Aβ RA-LTMRs, Aδ-LTMRs, and C-LTMRs and their TSC processes associated with awl/auchene and zigzag hair follicles are remarkably similar. On the other hand, Aβ RA-LTMR endings associated with guard hair follicles are distinct, having larger caliber lanceolate axonal endings and slightly thinner TSC processes compared to those of Aβ RA-, Aδ-, and C-LTMRs associated with awl/auchene and zigzag hair follicles. In addition, at awl/auchene and zigzag hair follicles, two or more axonal endings, which differ in mitochondrial abundance and belong to different LTMR subtypes, are often observed alternately arranged with three or more TSC processes, forming individual groups or 'units'. Thus, axon terminals of different LTMR subtypes can share the same set of TSC processes to form 'lanceolate complex units' at awl/auchene and zigzag hair follicles, which together comprise ~99% of trunk skin hairs. Such an arrangement was never observed at guard hairs; at guard hairs, each Aβ RA-LTMR lanceolate ending associates with three TSC processes to form stereotypical 'guard hair lanceolate complex units'. Thus, Aβ RA-LTMR lanceolate complexes at guard hair follicles are morphologically invariant and more precisely arranged into confined units than Aβ RA-, Aδ-, and C-LTMR complexes associated with awl/auchene and zigzag hairs, potentially endowing guard hair lanceolate complexes with higher fidelity or sensitivity to hair deflection.

The present study also defines the ultrastructural properties of lanceolate endings of the three physiologically defined LTMR subtypes that innervate hair follicles. A defining feature of Aβ RA-LTMR and Aδ-LTMR lanceolate endings is their highly abundant mitochondria. Neurons critically depend on mitochondrial ATP production for establishing membrane excitability and for calcium buffering (*Kann and Kovacs, 2007*), and emerging evidence suggests that mitochondria contribute to neuronal plasticity. An abundance of mitochondria in Aβ RA-LTMRs and Aδ-LTMRs terminals may be needed to support rapid conduction velocities, a distinguishing feature of these LTMR subtypes, compared to the slow-conducting C-LTMRs, whose lanceolate endings contain few mitochondria. We also observed several ultrastructural features of Aβ RA-, Aδ-, and C-LTMR lanceolate complexes that may be crucial to the development and maintenance of lanceolate complexes, and may also underlie their remarkable

sensitivity to hair deflection. Abundant small vesicles are found within the cytoplasm of axonal terminals of all LTMR subtypes. Studies by *Woo et al. (2012)* and *Banks et al. (2013)* suggest that these may be glutamate-containing vesicles that mediate signals from sensory neurons to TSCs. Consistent with this idea, Aβ RA-LTMRs and Aδ-LTMRs express VGluT1 and/or VGluT2, whereas C-LTMRs express VGluT3, suggesting that all LTMRs are glutamatergic (*Brumovsky et al., 2007*; *Seal et al., 2009*; *Woo et al., 2012*; *Banks et al., 2013*). Numerous pinocytotic vesicles were observed on both the inner and outer surfaces of TSC processes. These vesicle-like structures were noted previously and suggested to control ion flux around nerve fibers (*Yamamoto, 1966*). It is also reasonable to speculate that these invaginations serve as docking sites for axon fingers of longitudinal lanceolate endings (*Takahashi-Iwanaga, 2000*). Such an intercalating morphology increases the interaction surface area between TSC processes and LTMRs and may contribute to the mechanical sensitivity of LTMR endings. Most noteworthy, in our opinion, is the presence of intercellular processes that emanate from hemidesmosomes on outer root sheath epithelial cells of hair follicles, forming connections, or tethers, between hair follicle epithelial cells and Aβ RA-, Aδ-, and C-LTMR axon terminals and TSC processes. These filaments resemble anchoring filaments and anchoring fibrils, reported previously in other contexts to mediate dermal–epidermal adhesion (*Keene et al., 1987*; *Burgeson and Christiano, 1997*). At hair follicles, these filaments may contribute to assembly and maintenance of lanceolate ending-TSC-epithelial cell complexes. Intriguingly, the ~100-nm long filaments observed in the present study may be functionally analogous to the protein tethers described by Lewin et al. in cell culture experiments and proposed to link mechanosensitive ion channels in sensory neurons to the extracellular matrix (*Hu et al., 2010*). We speculate that the tether-like structures we observed at hair follicles in vivo may serve to facilitate or mediate excitation of lanceolate endings following hair deflection. Thus, we propose that 'epithelial cell–lanceolate complex tethers' serve a function that is analogous to the tip links that physically tether adjacent stereocilia of hair cells of the cochlea and vestibular apparatus and serve to open mechanically gated ion channels upon stereocilia movement. In the case of hair follicle epithelial cell–lanceolate complex tethers, we speculate that deflection of hair follicles would place strain on the tethers to activate mechanically gated ion channels situated on the membrane of the LTMR lanceolate ending thus leading to their depolarization. Interestingly, a recent study from Lewin et al. suggests that laminin 332, a major component of keratinocyte-derived extracellular matrix, prevents formation of protein tethers and inhibits rapidly adapting currents, further supporting the notion that extracellular matrices play a key role in mechanotransduction (*Chiang et al., 2011*). Potential molecular components of these tethers may include type VII collagen, a major component of anchoring fibrils, as well as other collagen proteins, laminins, and integrins, which could assemble into multimeric complexes to form tether structures (*Burgeson and Christiano, 1997*). Future studies will be needed to define the molecular components of these epithelial cell–lanceolate complex tethers. Once defined, tissue-specific ablation of key components, coupled with histological and physiological analyses, should reveal the contribution of these tethers for the assembly and maintenance of lanceolate complexes and/or their role in mechanotransduction during hair movement or deflection.

## TSCs support LTMR lanceolate ending integrity and are re-innervated following nerve injury

The intimate relationships between TSCs and longitudinal lanceolate endings of the three LTMR subtypes at each of the three hair follicle subtypes suggested to us an interdependence of LTMRs and TSCs during maintenance of lanceolate complexes. Indeed, we found that LTMR endings rapidly retract from hair follicles following TSC ablation. And yet, surprisingly, when LTMR endings are lost following a nerve cut, TSCs remain intact and intimately associated with hair follicles. In fact, TSCs of adult mice are remarkably stable in the absence of nerve, for at least several weeks or even months. This effect may be age-dependent, since we found that TSCs are less stable and often degenerate during the 2 weeks following skin denervation of neonates (data not shown). Thus, at neonatal ages, when lanceolate complexes are developing, LTMR axonal endings may be necessary for the maintenance of newly formed TSCs. Consistent with this, it was recently reported that TSCs associated with neonatal hair follicles are disorganized in *Wnt1Cre*-mediated VGLUT2 mutant mice, suggesting that neuronal glutamate signaling is required for proper organization of TSCs during lanceolate complex development (*Woo et al., 2012*). However, in the same study, blocking glutamatergic signaling through systemic administration of an NMDAR antagonist disrupts TSC structures at hair follicles in adult animals, which is different from our conclusion that sensory nerves are not required for

maintenance of TSCs at hair follicles in adult animals. One potential explanation for this discrepancy is that glutamatergic signaling from cells other than LTMRs is needed to maintain TSC processes. Nevertheless, our findings indicate that TSCs serve as anchors or scaffolds that maintain LTMR endings and lanceolate complex structure and integrity, whereas LTMR axonal endings of adult mice are not required for maintenance of TSCs.

It is remarkable that, within a few weeks following skin denervation, new axonal processes regenerate from severed dorsal cutaneous nerves and re-assemble with TSCs that had remained associated with denervated hair follicles. Moreover, the pattern of axonal innervation of hair follicles during regeneration exhibits similarities to that which is observed during development. In particular, guard hair follicles are the first to be re-innervated. At individual hair follicles, circumferential endings form first, surrounding TSC processes, and this is followed by gradual formation of longitudinal lanceolate endings during the next several days. This is comparable to the pattern of LTMR lanceolate complex formation observed during development of neonatal animals (data not shown) (*Peters et al., 2002*). Furthermore, LTMR-hair follicle subtype specificity appears conserved, at least for Aβ RA-LTMRs, during re-innervation. It is likely that the developmental and regeneration processes that govern LTMR lanceolate ending branching, extension, orientation, size, and TSC association have common mechanisms. Additionally, the precise role played by TSCs during the re-innervation of hair follicles is an intriguing question. During regeneration of the neuromuscular junction (NMJ) following axotomy, TSCs proliferate extensively and serve to guide motor axons to their destination by expressing some of the same cues that are used during development (*Reynolds and Woolf, 1992*; *Sanes and Lichtman, 1999*). TSCs at hair follicles may serve a similar role to attract specific subtypes of spouting LTMR axons to re-form lanceolate complexes. Future work will define the significance and molecular mechanisms underlying the intimate relationship between LTMR subtype longitudinal lanceolate endings and TSCs during formation, function, and regeneration of hair follicle-associated LTMR lanceolate complexes.

## Materials and methods

### Mouse lines

*Th^CreER*, *TrkB^tauEGFP*, *Npy2r-GFP* BAC transgenic, *Plp1^CreER*, *Rosa26^LSL-tdTomato*, *Rosa26-Confetti*, *Rosa26^GCaMP3*, and *Rosa26^LSL-hDTR* mouse lines, have been described previously (*Doerflinger et al., 2003*; *Gong et al., 2003*; *Buch et al., 2005*; *Rotolo et al., 2008*; *Madisen et al., 2010*; *Li et al., 2011*; *Schepers et al., 2012*; *Zariwala et al., 2012*). Generation of the *TrkA^f/f* (*TrkA^F592A*) mouse was described previously (*Chen et al., 2005*). In addition to the point mutation introduced into the ATP binding pocket, two Loxp sites were inserted into introns flanking exon 7 to 12 of the *TrkA* gene. *Wnt1Cre;TrkA^f/f* animals were generated to achieve ablation of the TrkA gene in cells of neural crest origin. The TrkA conditional knockout animals are viable and fertile, with no obvious behavioral abnormality, though behavioral assays assessing pain, temperature, and touch sensations are likely to reveal sensory defects in these TrkA conditional mutants.

### Immunohistochemistry of tissue sections

Protocols for immunohistochemistry were described previously (*Liu et al., 2007*; *Luo et al., 2009*; *Li et al., 2011*). Mice were anesthetized using $CO_2$ inhalation and transcardially perfused with PBS (pH 7.4, 4°C) followed by 4% paraformaldehyde (PFA) in PBS (pH 7.4, 4°C). DRGs and hairy skin were dissected from the perfused mice. DRGs were postfixed in PBS containing 4% PFA at 4°C for 1–2 hr. Hairy skin was postfixed with PBS containing 4% PFA at 4°C overnight. The tissues were cryoprotected in 30% sucrose in PBS at 4°C overnight, embedded in OCT (Tissue Tek) and frozen at −20°C. The tissues were sectioned at 20–30 µm using a cryostat. The sections on slides were dried at room temperature for 1 hr, and fixed with 4% PFA in PBS on ice for 15 min. The slides were washed with PBS containing 0.1% Triton X-100 (0.1% PBST) and blocked with 5% normal serum (goat or donkey) in 0.1% PBST at room temperature for 1 hr. The tissue sections were incubated with primary antibodies diluted in blocking solution at 4°C overnight. The next day, the sections were washed with 0.1% PBST, and incubated with secondary antibodies diluted in blocking solution at room temperature for 1 hr, washed again with 0.1% PBST, and mounted with fluoromount-G (Southern Biotech, Birmingham, AL).

The primary antibodies used for this study were: rabbit anti-CGRP (Immunostar, Hudson, WI, 24112, 1:1000), chicken anti-GFP (Invitrogen, A10262, 1:1000), rabbit anti-GFP (Invitrogen, Carlsbad, CA,

A11122, 1:1000), rabbit anti-NFH (Millipore, Billerica, MA, AB1982, 1:1000; Sigma, St. Louis, MO, N4142, 1:1000), chicken anti-NFH (Aves Labs, Tigard, OR, NFH, 1:1000), rabbit anti-S100 (DAKO, Denmark, Z0311, 1:1000), sheep anti-Tyrosine Hydroxylase (Millipore, AB1542, 1:400), rabbit anti-tuj1 (b-Tubulin) (Covance, Princeton, NJ, PRB-435P, 1:1000). The secondary antibodies used were: Alexa 488, 546 or 647 conjugated goat anti-chicken antibody, Alexa 488, 546 or 647 conjugated goat anti-rabbit antibodies, Alexa 546 conjugated donkey anti-sheep antibody. All secondary antibodies were purchased from Invitrogen.

## Whole-mount immunohistochemistry of hairy skin

Protocols for whole-mount skin immunostaining were described previously (*Li et al., 2011*). Back hairy skin from 3- to 10-week-old mice was treated with commercial hair remover, wiped clean with tissue paper and tape stripped until glistening. The skin was then dissected, cut into small pieces and fixed in 4% PFA in PBS at 4°C for 2 hr. The tissue was rinsed in PBS and then washed with PBS containing 0.3% Triton X-100 (0.3% PBST) every 30 min for 5–8 hr. Then, the skin was incubated with primary antibodies in 0.3% PBST containing 5% goat/donkey serum and 20% DMSO at room temperature for 3 to 5 days. After washing with 0.3% PBST every 30 min for 5–8 hr, the tissues were transferred to secondary antibodies in 0.3% PBST containing 5% goat/donkey serum and 20% DMSO and incubated at room temperature for 2 to 4 days. The tissues were then washed with 0.3% PBST every 30 min for 5–8 hr, dehydrated in 50% methanol for 5 min and 100% methanol for 20 min, three times, and lastly cleared in BABB (Benzyl Alcohol, sigma 402834; Benzyl Benzoate, sigma B-6630; 1:2) at room temperature for 20 min. To identify the types of hair follicles innervated by each LTMR class, similar whole-mount preparations of hairy skin were made without removing the hair. Using a confocal microscope, hair follicles were traced to the corresponding hair shafts. The number of rows of medulla cells in the hair shaft was counted to distinguish zigzag (1 row), awl/auchene (3 or 4 rows) and guard hairs (2 rows).

## Transmission electron microscopy

Adult mice were fixed by cardiac perfusion in 2% paraformaldehyde and 2.5% glutaraldehyde solution in 0.1 M phosphate buffer (pH 7.2). The back hairy skin was dissected on ice. Trunks of hairy skin with less than 20 hair follicles were carefully examined under dissection microscope to record the types and positions of hair follicles. Tissue samples were immersed in the same fixative overnight, treated with reduced 1% osmium tetroxide for 1 hr. To facilitate visualization of intercellular filaments between hair follicle epithelial cells and lanceolate complexes, some of the tissue samples were treated with 1% tannic acid for 5 min. All tissue samples were then stained with 2% uranyl acetate for 2 hr, dehydrated in graded ethanol and embedded in Epon. Toluidin blue-stained transverse sections of the hair follicles were made to determine the location of lanceolate endings and the positions for collecting ultrathin sections. Transverse thin sections were stained with uranyl acetate and lead citrate followed by examinations with Philips/FEI BioTwin CM120 transmission electron microscope at 80 kV.

## 4-hydroxytamoxifen (4-HT), tamoxifen and DTX injections

4-HT was dissolved in ethanol (10 mg/ml). For $Th^{CreER}$ animals, 100 µl (~1 mg) of 4-HT in ethanol was mixed with 200 µl of sunflower seed oil (Sigma), vortexed for 1 min and centrifuged under vacuum for 25 min to remove the ethanol. The 4-HT solution was delivered via oral gavage to $Th^{CreER}$ animals at P13, P14, and P15. For $Plp1^{CreER}$ animals, 2 mg of tamoxifen dissolved in 200 µl of sunflower seed oil was delivered via oral gavage to animals at P13 to P17; 0.01 to 0.03 mg of tamoxifen were used for sparse labeling.

DTX injections were performed as previously described (*Cavanaugh et al., 2009*). 40 ng/g of DTX (List Laboratories) was delivered by i.p. injections to mice at P21. Two treatments, separated by 72 hr, were given. Animals were sacrificed 2 to 3 weeks after the first injection.

## Dorsal cutaneous nerve axotomy

Animals were anaesthetized with Urethane (1500 mg/kg). A 2.5–3 cm midline incision was made in the dorsal skin under anesthesia. The dorsal cutaneous nerves were exposed under a dissection microscope. Dorsal cutaneous nerves (T5–L2) on the right side were cut with scissors. The skin was then closed with 5-0 sterile sutures. Back skin from the left side of each animal was used as control.

## Data analysis

Quantifications are presented as mean ± SEM. Statistical analyses were performed using unpaired Student's *t* tests. Significance was declared at $p < 0.05$ and was indicated by one star in the figures.

## Acknowledgements

We thank members of the Ginty laboratory for assistance and discussions throughout the course of this project, Baohan Pan for advice with surgical procedures, and Michael Caterina, Yin Liu, and Lawrence Schramm for thoughtful comments on the manuscript. DDG is an investigator of the Howard Hughes Medical Institute.

## Additional information

### Funding

| Funder | Grant reference number | Author |
|---|---|---|
| National Institutes of Health | R01NS34814 | David D Ginty |
| National Institutes of Health | R01DE022750 | David D Ginty |
| Howard Hughes Medical Institute | | David D Ginty |

The funder had no role in study design, data collection and interpretation, or the decision to submit the work for publication.

### Author contributions

LL, Conception and design, Acquisition of data, Analysis and interpretation of data, Drafting or revising the article; DDG, Conception and design, Analysis and interpretation of data, Drafting or revising the article

### Ethics

Animal experimentation: This study was done in accordance with the recommendations in the Guide for the Care and Use of Laboratory Animals of the National Institutes of Health. All of the animals were handled according to approved institutional animal care and use committee (IACUC) protocols of the Johns Hopkins University School of Medicine. The protocols were approved by the Animal Care and Use Committee of the Johns Hopkins University School of Medicine (Protocol Numbers: MO11M10). All surgery was performed under Urethane (1500 mg/kg) anesthesia, and every effort was made to minimize suffering.

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
