## [Decision Letter]

Thank you for submitting this very interesting manuscript, ‘The structure and organization of lanceolate mechanosensory complexes at mouse hair follicles‘, to *eLife*. Your article has been favorably evaluated by a Senior editor, a Reviewing editor, and 3 expert reviewers, one of whom, Gary Lewin, has agreed to reveal his identity. What follows is a consensus of their views, together with those of a member of the Board of Reviewing Editors.

This elegant study uses transgenic mouse models and ultrastructural analysis to identify anatomical features of mechanosensory endings around three mouse hair-follicle types. In particular, this manuscript examines the structural organization and interactions between low threshold mechanoreceptors (LTMRs) and terminal Schwann cells (TSCs) in mouse hairy skin. The authors use immunohistochemistry, EM and a diverse array of genetic tools that label different subsets of LTMRs or TSCs to understand how TSCs interact with distinct hair follicles and LTMR subtypes. The authors show that TSCs are organized in a mosaic or tiled fashion and interact with multiple subtypes of LTMR endings. In addition, the data show that conditional ablation of TSCs induces the retraction of LTMRs from hair follicles, while denervation has no gross effect on TSC organization. A particular strength of the manuscript is that the morphological and physiological organization of the terminal Schwann cell (TSC) has been investigated for the first time in detail. Overall the data are high quality and the manuscript is well written. However, a number of questions remain about some of the methodology, mouse lines used in the study, statistical comparisons, and interpretation of the data.

1) Quantitative analysis is an important strength of this study; however, specific conclusions require the reporting of statistics and quantification in a consistent manner. For example, based on qualitative data, the authors state ‘the organization of individually tiled TCSs and different LTMR endings can be visualized simply by immunostaining with glial markers.’ This method is then used in subsequent experiments. Providing quantification for the former observation would enhance confidence in the latter method. Also, in several instances, means are stated to be higher or lower without statistical comparisons. As a second example, the authors state that sparse labeling of TSCs shows multiple TSCs around each hair follicle but that their processes do not overlap with one another. The figure shows an example of one guard hair and one non-guard hair. It is not clear how many follicles were tested. The text states that 4 mice were used, but the actual number of follicles counted and the percentage of tiled TSCs would help illustrate the robustness of this organization that example images do not capture. As a third example, the mean cross sectional areas of axon terminals are given. The n is given as two mice but no information is given on the number of sections or number of lanceolates examined.

2) The TrkA mutant mouse. The authors state that in addition to C-LTMRs, the Aβ RA-LTMRs at hair follicles are also lost in TrkA conditional knockout animals (*Wnt1Cre;TrkA*^*f/f*^ ). This is surprising. Are the sensory cells still present? Figure 5 does not show a complete loss of fiber innervation. Has this mouse been characterized previously or is this the first report? What are the relative percentages of fibers in control versus mutant animals? Are the TSCs present with normal processes? Do the TSCs still interact normally with the Aδ-LTMRs in the mutant mouse? We realize that this mutant line is being used as a tool here, but some background on the phenotype is needed. The authors should mention the possibility that plasticity may occur here (TrkB positive D-hair afferents changing their innervation in the absence of other lanceolates?).

3) The authors suggest that reinnervation of hair follicles after denervation may be due to collateral sprouting. In the methods the authors state that the nerve was cut but give no further information. Normally cutting a peripheral nerve in situ will allow axons to regrow to the skin with relatively fast time course (in the mouse this is 4–6 weeks). Therefore the re-innervation that the authors see could be simply due to the reinnervation by the cut nerve rather than by collateral sprouting. If the authors had ligated the nerve this would favor the collateral sprouting model. The authors should clarify this issue. Were manipulations made to prevent regeneration of the cut nerve or not? The authors should exercise caution in their interpretation of this data. In the end it may not make a difference if follicles are re-innervated collaterally or by regrowing axons, but the majority of the literature spanning many decades is pretty clear about the fact that large diameter low threshold afferents tend not to produce much, if any collateral sprouting (experiments started by Henry Head in the 1900s and confirmed by many afterwards). If the authors really want to challenge this body of literature they need better experimental evidence and should provide it.

4) What happens to the nerve terminals after TSC ablation is unclear. How far do the fibers retract? What are the kinetics of the retraction? Additional data should be provided for the neuronal ablation section to better substantiate the claims. How many follicles are reinnervated? Does innervation of all subtypes of follicles re-form? This is important as a number of studies have proposed a role for neuronal signaling in Schwann cell development and maintenance. Indeed, [50] suggest a model of TSCs being dependent on neuronal glutamate signaling for development and maintenance of the follicular structure. These studies should be discussed.

5) Interpretation and discussion of data. It was not clear why the authors favor Col7a1 over other extracellular filaments Collagen 7a1 mutations are known to be causative in epidermolysis bullosa but mutations in other genes also cause this disease for example in genes encoding subunits laminin 332, which has been implicated in mechanotransduction (Chiang et al. Nat Neuro 2011). However, there is no reason to favor Col7a1 protein as opposed to numerous others (many large extracellular proteins are in the skin) as a tether for mechanotransduction. Staining for Col7a1 may be interesting but even positive light microscopy findings would not make this protein a better candidate than others. The manuscript would be strengthened by presenting a balanced view of candidates or by including additional experiments to support the Col7a1 model.

The proposal that ‘epithelial cell-lanceolate complex tethers’ are functionally analogous to tip links that open hair-cell transduction channels is highly speculative. It is equally possible that these tethers provide mechanical integrity without coupling to transduction channels, analogous to lateral links in hair bundles. The Discussion would be strengthened by delineating evidence needed to distinguish these possibilities.

The manuscript proposes that the precise arrangement of guard-hair lanceolates compared with awl/auchene and zigzag lanceolates might confer ‘higher fidelity or sensitivity to hair deflection.’ Physiological recordings are needed to support this model. In a previous study, these authors reported responses of A beta RA afferents (30). It would be interesting to know whether response properties different in afferents innervating guard versus awl/auchene hairs. An alternative interpretation is that these structural differences are simply a consequence of distinct development in primary (guard) hair follicles versus secondary follicles.

The Discussion includes a nice comparison with previous ultrastructural studies of lanceolate endings; however two recent, related studies are not discussed. Banks and colleagues (2013; J. Physiol. 15:2523-2540) reported that small clear-core vesicles in lanceolate terminals are enriched in glutamate. Examples shown by Banks et al. are quite similar to guard-hair lanceolate endings presented in Figure 4. Woo et al. (2012; Development 139:740-748) found that TSCs are disorganized around neonatal hair follicles in Wnt1-driven VGLUT2 knockout mice. As VGLUT2 is expressed in lanceolate terminals but not TSCs, these data imply that DRG-derived signals are required for proper development of TSCs. It would be helpful to discuss this study’s finding that TSC structure persists in denervated tissue in light of these previous reports.

[Editors' note: further clarifications were requested prior to acceptance, as described below.]

Thank you for resubmitting your work entitled “The structure and organization of lanceolate mechanosensory complexes at mouse hair follicles” for further consideration at *eLife*. Your revised article has been favorably evaluated by a Senior editor and a member of the Board of Reviewing Editors. Prior to acceptance we just have one additional point that we would like you to clarify:

In the original submission, it was unclear how C- and Aδ lanceolate endings were distinguished ultrastructurally in Figure 4–figure supplement 1 (now Figure 5). This is worth clarifying because the goal was to compare ultrastructure of ‘individually defined endings’. From the revision, it seems that the authors inferred these identities in wildtype mice based on the 1) the appearance of endings with two different mitochondrial densities in wild type mice, and 2) on the loss of endings with low mitochondrial density in the TrkA conditional knockout mice, which also lack C fibers. This indirect method relies on the assumption that the TrkA mutation has no effect on mitochondrial density in Aδ fibers.

The lack of an independent marker makes the graph in Figure 6—figure supplement 2 seem a bit circular. If I understand the experiment correctly, the authors quantified the mitochondrial density in wild type mice, and then assigned those with low density to the ‘C’ class and those with high density to the ‘Aδ’ or ‘Aβ’ class. Without an independent marker, it makes more sense to me to plot the data from each hair-follicle type as a single population with a bimodal distribution. By also plotting the distributions from TrkA CKO mice, one would expect to see a unimodal distribution whose range coincides with the upper ‘hump’ of the wild type bimodal distribution.

---

## [Author Response]

*1) Quantitative analysis is an important strength of this study; however, specific conclusions require the reporting of statistics and quantification in a consistent manner. For example, based on qualitative data, the authors state ‘the organization of individually tiled TCSs and different LTMR endings can be visualized simply by immunostaining with glial markers.’ This method is then used in subsequent experiments. Providing quantification for the former observation would enhance confidence in the latter method. Also, in several instances, means are stated to be higher or lower without statistical comparisons. As a second example, the authors state that sparse labeling of TSCs shows multiple TSCs around each hair follicle but that their processes do not overlap with one another. The figure shows an example of one guard hair and one non-guard hair. It is not clear how many follicles were tested. The text states that 4 mice were used, but the actual number of follicles counted and the percentage of tiled TSCs would help illustrate the robustness of this organization that example images do not capture. As a third example, the mean cross sectional areas of axon terminals are given. The n is given as two mice but no information is given on the number of sections or number of lanceolates examined*.

Mosaic labeling of TSCs was done using two *PLP*^*CreER*^*;Confetti* mice. In total, 109 non-guard hair follicles and 23 guard hair follicles were examined, all of which exhibited mosaic fluorescence labeling. 100% of these hair follicles showed tiled arrangements of TSCs at individual hair follicles. This information has been added to the legend for Figure 3.

The means, errors, sample sizes, and statistics regarding quantifications of the numbers and densities of TSCs, as well as cell body sizes, process widths and lengths of individual TSCs per hair follicle subtype were modified and included in the legend of Figure 2 of the revised manuscript.

Regarding the quantifications of cross sectional areas of axonal terminals, TSC processes and the exposed gaps of axonal terminals between TSC processes, numbers of sections and animals used for these measurements have been added to the Results section.

*2) The TrkA mutant mouse. The authors state that in addition to C-LTMRs, the Aβ RA-LTMRs at hair follicles are also lost in TrkA conditional knockout animals (*Wnt1Cre;TrkA^f/f^
*). This is surprising. Are the sensory cells still present?*
Figure 5
*does not show a complete loss of fiber innervation. Has this mouse been characterized previously or is this the first report? What are the relative percentages of fibers in control versus mutant animals? Are the TSCs present with normal processes? Do the TSCs still interact normally with the Aδ-LTMRs in the mutant mouse? We realize that this mutant line is being used as a tool here, but some background on the phenotype is needed. The authors should mention the possibility that plasticity may occur here (TrkB positive D-hair afferents changing their innervation in the absence of other lanceolates?)*.

This is the first report of experiments using the *Wnt1Cre;TrkA*^*f/f*^ conditional mutant mouse. We had previously reported the generation of the *TrkA*^*f/f*^ mouse (Chen et al., Neuron 2005). Additional description about the *Wnt1Cre;TrkA*^*f/f*^ mouse has been added to the Materials and methods section: “Generation of the *TrkAf/f* (*TrkAF592A*) mouse was described previously (9). In addition to the point mutation introduced into the ATP binding pocket, two Loxp sites were inserted into introns flanking exon 7 to 12 of the *TrkA* gene. *Wnt1Cre;TrkAf/f* animals were generated to achieve ablation of the TrkA gene in cells of neural crest origin. The TrkA conditional knockout animals are viable and fertile, with no obvious behavioral abnormality, though it is likely that behavioral assays assessing pain, temperature and touch sensations are likely to reveal sensory defects in these TrkA conditional mutants.”

Npy2r-GFP^+^ neurons are present in the TrkA conditional mutants. Images showing Npy2r-GFP neurons in DRG sections from control and TrkA conditional knockout animals have been added as Figure 6—figure supplement 1.

As shown in Figure 6, some Npy2r-GFP^+^, Aβ, RA-LTMR endings are occasionally observed in TrkA conditional knockout animals. For reasons we do not understand, these appear as free nerve endings that do not form longitudinal lanceolate endings at hair follicles.

So far it has not been technically feasible to quantify the relative percentages of lanceolate endings from different LTMR subtypes in awl/auchene and zigzag hair follicles in control animals due to the nature of mosaic genetic labeling. In TrkA conditional knockout animals, only Aδ-LTMRs remain at awl/auchene and zigzag hair follicles; neither Aβ RA-LTMR nor C-LTMR lanceolate endings can be found at hair follicles.

Whole-mount S100 immunostaining shows a slight decrease in number of TSCs at individual hair follicles from TrkA conditional knockout animals compared to control animals (Figure 11). Based on EM images, TSCs processes appear to be thin and hypotrophic at hair follicles from TrkA conditional knockout animals (Figure 11).Author response image 1.Characterization of TSCs at hair follicles of *Wnt1Cre;TrkAf/f* conditional mutant mouse.**A and A’.** Whole mount immunostaining with anti-S100 shows that numbers of TSCs at hair follicles from of *Wnt1Cre;TrkAf/f* conditional mutants (A’) appear slightly less than those in control animals (A). Scale bar, 20 μm.**B and B’.** Cross sections of lanceolate complexes at zigzag hair follicles from *Wnt1Cre;TrkAf/f* (B’) and control (B) mice. Axon terminals are pseudo-colored in green; TSC processes are colored in pink; hair follicle epithelial cells are colored in yellow. TSCs from TrkA conditional knockout hair follicles appear hypotrophic compared to control TSCs. Scale bar, 500 nm.**C.** Quantification of numbers of TSCs at individual guard, awl/auchene and zigzag hairs from control mice and TrkA conditional knockout mice. In control animals, there are 17.3 ± 0.7 TSCs at individual guard hair follicles (n=13 guard hairs), 4.7 ± 0.1 TSCs at awl/auchene hair follicles (n=34 awl/auchene hairs) and 3.8 ± 0.1 TSCs at zigzag hair follicles (n=54 zigzag hairs). In TrkA conditional knockout animas, there are 9.7 ± 0.7 TSCs at individual guard hair follicles (n=15 guard hairs), 3.0 ± 0.1 TSCs at awl/auchene hair follicles (n=44 awl/auchene hairs) and 2.6 ± 0.1 TSCs at zigzag hair follicles (n=83 zigzag hairs) (p<0.001 for comparisons between all three hair follicle subtypes).

The general morphology of Aδ-LTMRs and their anatomical relationships with TSC processes appear normal based on TrkB^tauEGFP+^ immunostaining and EM analyses (Figure 6). However, further physiological recordings from TrkB^tauEGFP+^ neurons from ex viso skin-nerve or in vivo preparations would be needed to determine whether Aδ-LTMR lanceolate endings respond normally to mechanical stimulation in the absence of Aβ, RA-LTMRs and C-LTMRs.

*3) The authors suggest that reinnervation of hair follicles after denervation may be due to collateral sprouting. In the methods the authors state that the nerve was cut but give no further information. Normally cutting a peripheral nerve in situ will allow axons to regrow to the skin with relatively fast time course (in the mouse this is 4-6 weeks). Therefore the re-innervation that the authors see could be simply due to the reinnervation by the cut nerve rather than by collateral sprouting. If the authors had ligated the nerve this would favor the collateral sprouting model. The authors should clarify this issue. Were manipulations made to prevent regeneration of the cut nerve or not? The authors should exercise caution in their interpretation of this data. In the end it may not make a difference if follicles are re-innervated collaterally or by regrowing axons, but the majority of the literature spanning many decades is pretty clear about the fact that large diameter low threshold afferents tend not to produce much, if any collateral sprouting (experiments started by Henry Head in the 1900s and confirmed by many afterwards). If the authors really want to challenge this body of literature they need better experimental evidence and should provide it*.

Thank you for pointing this out. In our experiments, dorsal cutaneous nerves (T5-L2) were severed, thereby denervating skin supplied by branches of both medial and lateral dorsal cutaneous nerves. No ligations were performed to prevent regeneration of the cut nerves.

Reinnervation was most prominent at the edges of the denervated field during the early period following axotomy. This may be explained by limited sprouting from the intact T4 and L3 dorsal cutaneous nerves within their territory. Lightly myelinated and myelinated nociceptive axons that are labeled by NFH immune-labeling may contribute more to the collateral sprouting we observed. Based on the literature regarding peripheral nerve regeneration (Jackson et al., Journal of Comparative Neurology 1984), and as indicated by the reviewers, the majority of reinnervation is likely the result of regeneration of the cut nerves. Indeed, the delayed reinnervation at the center of the denervated field may be due to its further distance from the injured sites of the dorsal cutaneous nerves supplying this region. The Results section has been modified accordingly.

*4) What happens to the nerve terminals after TSC ablation is unclear. How far do the fibers retract? What are the kinetics of the retraction? Additional data should be provided for the neuronal ablation section to better substantiate the claims. How many follicles are reinnervated? Does innervation of all subtypes of follicles re-form? This is important as a number of studies have proposed a role for neuronal signaling in Schwann cell development and maintenance. Indeed,*
[50]
*suggest a model of TSCs being dependent on neuronal glutamate signaling for development and maintenance of the follicular structure. These studies should be discussed*.

Longitudinal lanceolate endings are lost upon TSC ablation. It is unclear how far each type of LTMR fiber retracts based on generic neuronal marker staining, such as Tuj1 and NFH. As shown by NFH staining in Figure 12, in the absence of TSCs, many fibers remain in the dermis, while no terminals forming lanceolate endings structures remain at the hair follicles.Author response image 2.Characterization of cutaneous nerve upon TSC ablation.Immunostaining with anti-S100 and anti-NFH was performed on skin sections from *PLP*^*CreER*^ (A) and *PLP*^*CreER*^*;Rosa 26*^*hDTR*^ mice (A’) to label TSCs and myelinated cutaneous nerves. In the absence of TSCs (A’), many fibers remain in the dermis, while no terminals forming lanceolate endings structures remain at the hair follicles. Scale bar, 100 μm.

The disappearance of longitudinal lanceolate endings is coincident with degeneration of TSCs; circumferential lanceolate endings are more resilient and remain in the absence of TSCs. This point has been added to the revised manuscript and in Figure 8—figure supplement 1.

In the skin denervation experiments (“neuronal ablation section”), by five weeks after denervation, approximately 90% of hair follicles exhibited at least some re-innervation. Lanceolate complexes at guard hairs have considerably more advanced recovery compared to non-guard hair follicles. Most non-guard hair follicles, especially zigzag hair follicles, display only sparse circumferential endings at this stage. This statement has been added to the Results section.

[50] suggest that neuron-derived glutamatergic signaling is required for proper organization of TSCs during development of lanceolate complexes. Consistent with that, we observed that when denervation was performed using neonatal animals, TSCs were less stable and often degenerate during the two weeks following skin denervation (data not shown). However, in the same study, blocking glutamatergic signaling by systemic treatment with an NMDAR antagonist disrupts TSC structures at hair follicles in adult animals. This is different from our conclusion that sensory nerves are not required for maintaining TSCs at hair follicles in adult animals. One potential explanation for this discrepancy is that glutamatergic signaling from cells other than lanceolate endings is required to maintain TSC processes. The Discussion section has been modified to incorporate this point.

*5) Interpretation and discussion of data. It was not clear why the authors favor Col7a1 over other extracellular filaments Collagen 7a1 mutations are known to be causative in epidermolysis bullosa but mutations in other genes also cause this disease for example in genes encoding subunits laminin 332, which has been implicated in mechanotransduction (Chiang et al. Nat Neuro 2011). However, there is no reason to favor Col7a1 protein as opposed to numerous others (many large extracellular proteins are in the skin) as a tether for mechanotransduction. Staining for Col7a1 may be interesting but even positive light microscopy findings would not make this protein a better candidate than others. The manuscript would be strengthened by presenting a balanced view of candidates or by including additional experiments to support the Col7a1 model*.

Thank you for pointing this out. The following statement has been added to the Discussion section: “Potential molecular components of these tethers may include type VII collagen, a major component of anchoring fibrils, as well as other collagen proteins, laminins, and integrins, which could assemble into multimeric complexes to form tether structures (5). Future studies will be needed to define the molecular components of these epithelial cell-lanceolate complex tethers. Once defined, tissue-specific ablation of key components coupled with histological and physiological analyses should reveal the contribution of these tethers for the assembly and maintenance of lanceolate complexes and/or their role in mechanotransduction during hair movement or deflection.”

*The proposal that ‘epithelial cell-lanceolate complex tethers’ are functionally analogous to tip links that open hair-cell transduction channels is highly speculative. It is equally possible that these tethers provide mechanical integrity without coupling to transduction channels, analogous to lateral links in hair bundles. The Discussion would be strengthened by delineating evidence needed to distinguish these possibilities*.

There are indeed at least two potential functions of the putative tethers: they may contribute to assembly of lanceolate complexes and maintenance of their association with hair follicles; and/or serve to facilitate or mediate excitation of lanceolate endings during hair deflection. These two possibilities are equally possible and not mutually exclusive. Future in vivo studies will be needed to define the molecular components of these tether structures. Once defined, tissue- specific ablation of key molecular components coupled with histological and physiological analyses would be required to assess the contributions of these epithelial cell-lanceolate complex tethers to assembly and maintenance of lanceolate complexes and/or their role in mechanotransduction during hair movement or deflection. The Discussion section has been modified accordingly.

*The manuscript proposes that the precise arrangement of guard-hair lanceolates compared with awl/auchene and zigzag lanceolates might confer ‘higher fidelity or sensitivity to hair deflection.’ Physiological recordings are needed to support this model. In a previous study, these authors reported responses of A beta RA afferents (*[30]*). It would be interesting to know whether response properties different in afferents innervating guard versus awl/auchene hairs. An alternative interpretation is that these structural differences are simply a consequence of distinct development in primary (guard) hair follicles versus secondary follicles*.

This is a very interestingly question. However, currently, we know of no genetic tools that will allow us to distinguish Aβ RA-LTMRs that innervate guard hairs vs awl/auchene hairs. We do aim to find useful mouse molecular genetic tools that will allow us to address this interesting issue.

*The Discussion includes a nice comparison with previous ultrastructural studies of lanceolate endings; however two recent, related studies are not discussed. Banks and colleagues (2013; J. Physiol. 15:2523-2540) reported that small clear-core vesicles in lanceolate terminals are enriched in glutamate. Examples shown by Banks et al. are quite similar to guard-hair lanceolate endings presented in*
Figure 4*. Woo et al. (2012; Development 139:740-748) found that TSCs are disorganized around neonatal hair follicles in Wnt1-driven VGLUT2 knockout mice. As VGLUT2 is expressed in lanceolate terminals but not TSCs, these data imply that DRG-derived signals are required for proper development of TSCs. It would be helpful to discuss this study’s finding that TSC structure persists in denervated tissue in light of these previous reports*.

Thank you for pointing this out. We have added a few sentences to the Discussion: “Abundant small vesicles are found within the cytoplasm of axonal terminals of all LTMR subtypes. Studies by [50] and [2] suggest that these may be glutamate containing vesicles that mediate signals from sensory neurons to TSCs. Consistent with this idea, Aβ RA-LTMRs and Aδ-LTMRs express VGluT1 and/or VGluT2, while C- LTMRs express VGluT3, suggesting that all LTMRs are glutamatergic (3; 47; 50; 2).”

The study by [50] has also been discussed under responses to major point 4 (please see above) and in the Discussion section of the revised manuscript.

[Editors' note: further clarifications were requested prior to acceptance, as described below.]

*In the original submission, it was unclear how C- and Aδ lanceolate endings were distinguished ultrastructurally in Figure 4*–*figure supplement 1 (now*
Figure 5*). This is worth clarifying because the goal was to compare ultrastructure of ‘individually defined endings’. From the revision, it seems that the authors inferred these identities in wildtype mice based on the 1) the appearance of endings with two different mitochondrial densities in wild type mice, and 2) on the loss of endings with low mitochondrial density in the TrkA conditional knockout mice, which also lack C fibers. This indirect method relies on the assumption that the TrkA mutation has no effect on mitochondrial density in Aδ fibers*.

Yes, we infer the identity of C-LTMRs, Aδ-LTMR and Aβ-LTMR endings in our EM analysis based upon the finding that Aβ-LTMR endings exhibit high mitochondrial densities around Guard hairs, which only have Aβ-LTMR endings, and mitochondrial densities around wild-type and TrkA knockout Zigzag hairs, which have both C-LTMR and Aδ-LTMR or only Aδ-LTMR lanceolate endings, respectively. As the Reviewing Editor points out, our method relies on the assumption that mitochondrial density and the electron dense nature of Aδ-LTMR endings does not undergo a dramatic change in *TrkA* conditional knockout mice. We think this is a reasonable assumption because Aδ-LTMR LTMRs do not express TrkA. We have modified the sentence on page 9 to read: Assuming that the *TrkA* mutation has no effect on mitochondria density in Aδ- LTMR axonal terminals, this finding indicates that the lanceolate endings with few mitochondria belong to C-LTMRs while those with abundant mitochondria belong to Aδ-LTMRs or Aβ RA- LTMRs (Figure 5; Figure 6).

*The lack of an independent marker makes the graph in*
Figure 6—figure supplement 2
*seem a bit circular. If I understand the experiment correctly, the authors quantified the mitochondrial density in wild type mice, and then assigned those with low density to the ‘C’ class and those with high density to the ‘Aδ’ or ‘Aβ’ class. Without an independent marker, it makes more sense to me to plot the data from each hair-follicle type as a single population with a bimodal distribution. By also plotting the distributions from TrkA CKO mice, one would expect to see a unimodal distribution whose range coincides with the upper ‘hump’ of the wild type bimodal distribution*.

We have taken this advice and replotted the data as distribution plots. Indeed, as predicted, there is a clear bimodal distribution of mitochondrial density in LTMR endings associated with Zigzag and Awl/auchene hairs of wild-type mice while, in contrast, there is a unimodal distribution corresponding to the upper “hump” in endings associated with Zigzag hairs. Thus, endings with few mitochondria are entirely missing in the TrkA conditional mutants. This is the basis of our argument that endings with few mitochondria belong to C-LTMRs, which are missing in TrkA mutants, whereas those with abundant mitochondria belong to Aδ-LTMRs. A new figure showing these distribution data now replaces the older version of Figure 6—figure supplement 2.